# ACTIVE SIDE CHANNEL ANALYSIS FOR CROSS DEVICE ATTACK

## ABSTRACT

Side Channel Analysis (SCA) exploits relationships between physical signals of a device and its actual computation to extract sensitive information, causing serious threat to privacy and security. Among various approaches, Deep Learning-based profiling attacks (DL-SCA) have recently emerged as one of the most powerful methods due to their ability to fully characterize the target devices. However, they suffer from major drawbacks including huge data consumption and lack of portability across different target devices. This paper introduces Active SCA (ActSCA), a *unique* and *generic* framework for boosting performance of any base DL-SCA model. ActSCA fundamentally differs to existing research as follows. Firstly, rather than relying on large training data in the profiling stage, it *actively* selects subsets of training data and *iteratively* refine the model to avoid overfitting, thus enhancing performance. Secondly, in the attack stage, ActSCA *exploits* existing training data pool from profiling devices to construct *separate* attack models for different target devices *without requiring any training data from the attacking devices* as is the case in other existing methods by using only *few unlabeled SCA traces* collected during the attacking phase to guide the model adaptation process. These make ActSCA a highly *portable* and *practical* attack method. We demonstrate its performances on both Post Quantum (PQC) Kyber and non-PQC Advanced Encryption Standard (AES) cryptography using power leakage to retrieve secret keys as case studies. ActSCA significantly improves the performances of all employed base models and outperforms all recent approaches like CNNC, MDMSD, ZMUV, MMD, ADA in terms of mean rank and top-$k$ accuracy.

## 1 INTRODUCTION

Due to the prevalence of IoT, it has become very challenging to ensure the security of embedded devices as they are easily accessible to adversaries. Although security primitives are resistant to mathematical attacks, their cryptographic implementations may still leak sensitive information. One such attack is SCA, where an attacker captures signals emanating from a victim device and correlates these with information from the device functionality to reveal se-

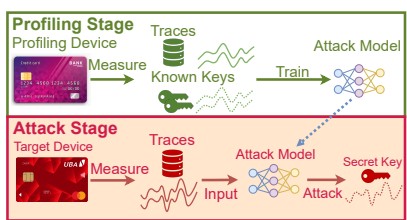

Figure 1: DL-based profiling attacks.

cret information. These signals can take many forms, including power consumption (Bhasin et al., 2020), EM emanations (Agrawal et al., 2003), time to execute (Coppens et al., 2009), and others (e.g., (Lipp et al., 2020; Zhao & Suh, 2018)). Among them, power and EM side channels are often the target. Both leakages are caused by the electrical switching of internal gates of devices, which consumes power and release EM emanations. Hence, different operations or data values have different impacts on these measurements due to state changes. Depending on the form of the side channel, the adversary can target probing many parts inside the cryptographic algorithm especially for secret key recovery. SCA can be broadly classified into two categories: profiling attacks and non-profiling attacks. In a non-profiling attack, the attacker only observes the physical leakages (e.g., power, or EM) on the target device (Mangard, 2002; Lathrop, 2020). In a profiling attack (cf. Figure 1), the attacker can access to a physical mitigated version of the target device, e.g. (Dubrova et al., 2023).

In SCA, profiled attacks are widely recognized as one of the strongest attacking method due to their ability to fully characterized the profiling devices (Bhasin et al., 2020). While there are traditional

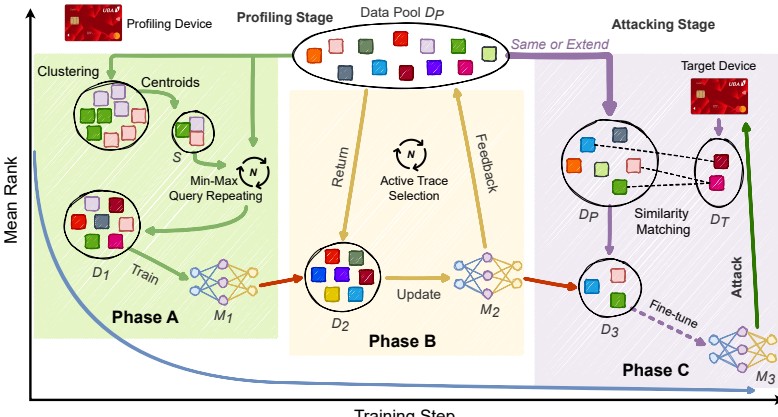

Figure 2: Overviews of our Active Side Channel Analysis (ActSCA) framework with 3 Phases: (A) Data representation selection, (B) Active model updating, and (C) Cross device adjustment.

approaches for profiling attack such as Template Attack (Chari et al., 2003) and Stochastic Model (Doget et al., 2011), their performance and usage are limited (Jin et al., 2020) and are surpassed by traditional ML methods like SVM (Heuser & Zohner, 2012) or LS-SVM (Hospodar et al., 2011). Recently, Deep Learning (DL) has emerged in profiling-based SCA as one of the most effective approaches due to their ability to fully characterize the profiled devices (Cagli et al., 2017; Maghrebi et al., 2016). The employed DL architectures are also diverse such as MLPs (Martinasek et al. 2014), RNNs (Dubrova et al., 2023), Transformers (Hajra et al., 2024), LSTM (Ahmed et al. 2023), and CNNs (Zaid et al., 2020). The target cryptography scheme also varies such as NIST PQC candidates like SABER (D'Anvers et al.), FrodoKEM (Alkim et al.), or CRYSTAL-Kyber (Hoang et al., 2024) and non-PQC like Advanced Encryption Standard (AES) (Emmanuel et al., 2018).

However, all the above approaches require large amounts of data to train their DL models. This increases computational cost without necessarily improving performance, especially when signals are overlapping or distorted (Cagli et al., 2017). This raises an interesting question: *can we improve the model performance by carefully exploiting a subset of data rather than asking for more data?* However, none of existing DL-SCA techniques aim to solve this issue, to the best of our knowledge. Additionally, *most DL-SCA studies use a single device for both profiling and attacking—an unrealistic setting*, as even similar devices can produce significantly different signals due to manufacturing variations and measurement setups (Das et al., 2019). Hence, they tend to not perform well in reality (Bhasin et al., 2020). This phenomenon is called *portability* and has become an emerging research topic in SCA recently. Some works aim to enhance the diversity of training data via many profiled devices such as (Rioja et al., 2020). Other methods like (Wu et al., 2023) try to reduce model overfitting by removing DL model's layers. A few works such as (Yu et al., 2021) apply transfer learning to reduce required traces on the target device. However, these methods still depend on *large amount of data* from the target device for training models (Yu et al., 2023; 2021) or overlook device-specific characteristics during attacks (Wu et al., 2023), which limits their practicality.

**Our Contributions.** We introduce Active SCA (**ActSCA**), a *unique* and *generic* approach to cope with above challenges. Compared to existing works, ActSCA fundamentally differs as follows.

First, ActSCA enables the DL model to actively select training data that is most beneficial to its performance. Initially, it identifies a small, representative subset of samples in the hope that these samples can cover all of the data characteristics, thus creating a model with adequate performance but less overfitting. After that, it *iteratively* evaluates all training samples to choose sets of important ones to update the model and enhance the performance. As shown in Section 3.3, ActSCA can reach better overall performance compared to full data training. Unused training data will be retained in the training pool and used as or *unseen* data during the attack stage to construct a new unique model for each specific target device from the base one. This reduces the bias of the new models towards the base one due to data diversity, thus increasing their generalization and overall performance.

Second, we propose a *unique* strategy for attacking cross devices, directly tackling the *portability* problem. Rather than building models based on training data from the target device like existing works (which is impractical) (Yu et al., 2023; 2021), ActSCA *does not require any training data from the target device*. Instead, during the attacking stage, it exploits the pool of profiling traces

from the profiling devices to create a new *specific* model for the target device by finding traces that are most similar to the target traces as inputs to refine the base model obtained from the previous step above. This *reverse* training scheme: (i) allows label-free target traces, (ii) requires fewer target traces, (iii) targets device-specific models, (iv) reduces reliance on multiple profiling devices, and (v) enables the integration of additional data into the existing pool without full retraining—making ActSCA a highly *portable* and *practical* approach compared to existing works (c.f. Section 3.1).

**Case studies.** As a *generic* framework, ActSCA can be used on different existing DL-SCA models. We demonstrate its performance in attacking cryptographic algorithms using power traces to retrieve secret keys as case studies including non-PQC AES (Emmanuel et al., 2018) and especially PQC CRYSTAL-Kyber (Hoang et al., 2024). Extensive experiments are conducted on multiple devices, different DL architectures (e.g. MLP, CNN and Transformer), and a set of 300 different test keys (going beyond single-key attacks like existing works (Hoang et al., 2024)). ActSCA significantly boosts the overall performance of all employed DL models for both same and cross device attacks, and outperforms all recent approaches such as CNNC (Hoang et al., 2024), MDMSD (Wu et al., 2023), ZMUV (Montminy et al., 2013), MMD (Cao et al., 2021), and ADA (Wang et al., 2023).

## 2    OUR PROPOSED METHOD ACTSCA

As depicted in Figure 1, a typical profiled SCA attack consists of 2 stages: *Profiling* and *Attacking*. The Profiling stage is conducted on a profiling device to obtain an attacker that models the dependency of SCA traces and secret keys. In the Attacking stage, SCA traces are measured on the target device and used as inputs for the trained attack model from the Profiling stage to infer secret keys.

**Problem formulation.** We focus on *cross-device profiling attacks*, in which we have a *profiling device* ($PD$) to collect training data to train a DL attacking model $M$ and retrieve secret keys of another *target device* ($TD$) under a cryptographic algorithm $\mathcal{C}$ using power leakage. Let $D_P = \{(\mathbf{x}_i, y_i)\}_{i=1}^{N_P}$ be a training data pool collected during the *profiling stage*, where each power trace $\mathbf{x}_i \in \mathbb{R}^{d_{in}}$ is captured during the execution of $\mathcal{C}$ on *PD*, with corresponding key label $y_i \in \mathcal{K}$, where $\mathcal{K}$ is the set of all secret keys. $N_P$ can be arbitrary large since ciphertexts can be freely generated (Emmanuel et al., 2018). Let $D_T = \{(\mathbf{x}_i', k)\}_{i=1}^{N_T}$ be a small set of traces collected from $TD$ in the *attacking stage*, where $k \in \mathcal{K}$ be the associated secret key to be retrieved by $M$ (i.e., $k$ is unknown). Different to existing works such as (Yu et al., 2021; 2023), we adopt a more practical setting in which $N_T$ is very small and no additional labeled training data is obtained from $TD$. In DL-SCA, the attack model $M$ is generally a classifier $M_\theta : \mathbb{R}^{d_{in}} \to \mathbb{R}^{d_{out}}$, where $d_{out} = |\mathcal{K}|$ is the number of classes corresponding to all possible secret key values and $\theta$ is the set of model parameters. Let $\sigma$ be the softmax function and $\hat{y} = \sigma(M_\theta(\mathbf{x}))$ be the predicted class distribution of a trace $x$. Our objective is to optimise the model $M_\theta$ on the training pool $D_P$ to minimize rank $r_{N_T}(k)$ (Emmanuel et al., 2018) of true key $k$ over $D_T$:

$$r_{N_T}(k) = \sum_{y \in \mathcal{K}} \mathbb{1}_{s_{N_T}(y) \geq s_{N_T}(k)} \tag{1}$$

where $r_{N_T}(k) = 1$ indicates correct key $k$ has the highest score and $s_{N_T}(k) = \sum_{i=1}^{N_T} \log{[\hat{y}_i[k]]}$ is the predictive score for a candidate key $k$ over $D_T$. Specifically, ActSCA aims to: (i) improve performance with fewer training samples from $D_P$; (ii) enable attacks with very small number of unlabeled target traces ($N_T \geq 1$); and (iii) generate device-specific models $M_{\theta_i}$ (or simply $M_i$) for better *portability* when attacking each specific target device $TD_i$. We use the terms *profiling* (or *same*) and *attacking* (or *cross*) to denote the profiling and target devices.

**Our general approach.** Figure 2 illustrates our ActSCA framework, a novel method for cross-device profiling attacks that significantly enhances base attack performance and generalization across target devices. In the ***profiling stage***, rather than training on the full training pool $D_P$ like existing works (Cagli et al., 2017; Hoang et al., 2024; Zaid et al., 2020), thus (possibly) suffering from distored signals and overfitting (Cagli et al., 2017), ActSCA *iteratively* and *actively* selects informative data subsets to improve its performance from both *data* and *model perspectives* in Phases A and B, respectively. Specifically, Phase A aims to select a set $D_1$ of representative samples that effectively capture important characteristics of $D_P$ to train an initial model $M_1$ with adequate performance while being less overfitted to $D_P$. Then, Phase B iteratively refines $M_1$ using small sets of samples $D_2$ selected from $D_P$ by the model itself, resulting in a more effective model $M_2$, which

can be used to attack target devices ($TDs$) directly. Moreover, in the **attacking stage**, we introduce Phase C, a *unique* strategy for further enhancing portability, that does not need any training data from $TD$ or ignores its characteristics as in the case of existing works (Yu et al., 2021; 2023; Wu et al., 2023). For each $TD$, ActSCA leverages *similarity* between profiling and target traces from both *feature* and *prediction viewpoints* to adapt the base model $M_2$ into a device-specific model $M_3$ using a subset of traces $D_3 \in D_P$, enabling specific-cross-device attacks with minimal target data.

## 2.1 PHASE A - DATA REPRESENTATION SELECTION

The key idea of Phase A is to choose a compact yet informative subset $D_1 \subset D_P$ to train model $M_1$, which can effectively represent the essential structure of the data while avoiding overfitting to $D_P$. To achieve this, the subset $D_1$ must be both *representative* (capturing important data characteristics) and *diverse* (covering broad regions of the input space). To do so, we employed a two-step approach that finds the representative samples as the initial set first and then expands the diversity of this set by a unique sampling strategy (cf. Algorithm 3 in Appendices for pseudo-codes).

**Data clustering and sample selection.** First, we employ $k$-medoids clustering (Schubert & Rousseeuw, 2021) to group all traces $T(D_P) = \{\mathbf{x}_1, \ldots, \mathbf{x}_n\}$ of $D_P$ into a set of $k$ disjoint clusters $C = \{C_1, \ldots, C_k\}$ under the distance metric $d(\mathbf{x}_i, \mathbf{x}_j) = ||\mathbf{x}_i - \mathbf{x}_j||^2$. Then we select a subset $S = \{(\mathbf{s}_i, l(\mathbf{s}_i)) | \mathbf{s}_i \in T(D_P) \wedge i \in [1, k]\}$, where $\mathbf{s}_i = \arg\min_{\mathbf{x} \in C_i} \sum_{\mathbf{y} \in C_i} d(\mathbf{x}, \mathbf{y})$ is the centroid of cluster $C_i$ together with its key label $l(\mathbf{s}_i) \in \mathcal{K}$. Since each centroid represents its cluster, $S$ contains most representative samples of $D_P$. Initially, we have $D_1 = S$. Note that we can replace $k$-medoids by other clustering methods like $k$-means as shown in Section F in the Appendices.

**Balanced max-min sampling (BMMS).** Second, we proposed BMMS, a special iterative sampling strategy to select more diverse samples into the representative set $D_1$. The key idea is to iteratively choosing a new sample $\mathbf{x} \in D_P \backslash D_1$ that are most different to its nearest sample $\mathbf{p} \in D_1$ via the distance metric $d$ to expand $D_1$ for diversity. Additionally, we try to balance the label distribution of $D_1$ during sampling to mitigate class imbalance, which can degrade performance (He & Garcia, 2009). The label distribution in $D_1$ is preserved by favoring samples with rare labels, determined by the cardinality of the per-class subset $D^{y_i} = \{(\mathbf{x}, y) \in D_1 : y = y_i\}, \forall y_i \in \mathcal{K}$. The BMMS process iteratively selects a sample $(\mathbf{x}_b, y_b) \in D_P \backslash D_1$ that maximizes the joint objective function:

$$J_1(\mathbf{x}_i, y_i) = \min_{(\mathbf{x}_j, y_j) \in D_1} \gamma \exp\Big(\text{norm}(d(\mathbf{x}_i, \mathbf{x}_j)) - 1\Big) + (1 - \gamma)\Big(1 - \exp(|D^{y_i}|/|D_1| - 1)\Big) \quad (2)$$

where hyperparameter $\gamma \in [0, 1]$ controls the trade-off between the two components: *Distance-based Diversity* (first term) and *Label Balance* (second term) and $norm()$ is the normalization of the distance $d$ into $[0, 1]$. BMMS is repeated until we reach a predefined number of samples $\delta$ (i.e. $|D_1| = \delta$) to train an initial model $M_1$.

**Theoretical analysis.** Let $l(\mathbf{x}_i, y_i; M_1)$ be the loss of sample $(\mathbf{x}_i, y_i)$ with respect to model $M_1$ on $S$, the Subset Loss reflects the empirical differences in loss between the chosen subset $S$ and $D_P$.

**Definition 1 (Subset Loss)** *Given the full data pool $D_P = \{(\boldsymbol{x}_i, y_i)\}_{i=1}^{N_P}$ and a chosen subset $S \subset D_P$ with size $N_S = k$. The Subset loss between fullset $D_P$ and subset $S$ is defined as follows:*

$$\mathcal{L}_{subset}(D_P, S) = \left| 1/N_P \sum_{(\boldsymbol{x}_i, y_i) \in D_P} l(\boldsymbol{x}_i, y_i; M_1) - 1/N_S \sum_{(\boldsymbol{x}_j, y_j) \in S} l(\boldsymbol{x}_j, y_j; M_1) \right| \quad (3)$$

We formally bound this loss in Theorem 1, showing that Subset loss is bounded by the maximum cluster radius $R$, plus a a statistical error term that vanishes as data size increase $N_P \to \infty$. As $N_P$ is typically large, this theorem guarantees that training on the subset $S$ achieves similar generalization as using the full dataset while mitigating overfitting (c.f. Appendix I for proof).

**Theorem 1** *Given the full data pool $D_P = \{(\boldsymbol{x}_i, y_i)\}_{i=1}^{N_P}$ with total $|\mathcal{K}|$ classes and a chosen subset $S = \{(\boldsymbol{x}_j, y_j)\}_{j=1}^{N_S} \subset D_P$. If loss function $l(\cdot, y, \theta)$ is $K^l$-Lipschitz continuous for all $y, \theta$, bounded by $L$ and model $M_1$ is $K^\mu$-Lipschitz, $R$ is the largest radius of all clusters of $k$-medoids, $l(\boldsymbol{x}_j, y_j; M_1) = 0$ for all $(\boldsymbol{x}_j, y_j) \in S$ then with probability at least $1 - \xi$, we have the inequality:*

$$\left| \frac{1}{N_P} \sum_{(\boldsymbol{x}_i, y_i) \in D_P} l(\boldsymbol{x}_i, y_i; M_1) - \frac{1}{N_s} \sum_{(\boldsymbol{x}_j, y_j) \in S} l(\boldsymbol{x}_j, y_j; M_1) \right| \le R(K^l + K^\mu L|\mathcal{K}|) + \sqrt{\frac{L^2 \log(1/\xi)}{2N_P}}$$

## 2.2 PHASE B - ACTIVE MODEL UPDATING

While Phase A selects training data from a *data perspective* by prioritizing diversity and representativeness across the input space, Phase B shifts to a *model perspective*, where the model $M_2$ (initialized from $M_1$) *actively* identifies and learns from samples that are most beneficial to its own improvement. This novel *iterative self-improvement* approach differs from prior works, which typically rely on static datasets and ignore the model's learning behaviors during training. Phase B also includes two steps as described below (cf. Algorithm 4 in Appendices for pseudo-codes).

**Active data selection.** Rather than training on the full dataset indiscriminately, the model selectively focuses on *informative samples*, which can potentially help to reduce generalization error and improve model confidence (Raj & Bach, 2022). Given a model $M$ and softmax function $\sigma$, the model-perpspective informativeness of a sample $\mathbf{x}$ with label $y$, denoted as $H(\mathbf{x})$, is defined as:

$$H(\mathbf{x}) = -\sum_{k=1}^{|\mathcal{K}|} \sigma(M((y=k|\mathbf{x}))) \log \sigma(M((y=k|\mathbf{x}))) \tag{4}$$

Maximum entropy $H(\mathbf{x})$ is achieved when the predicted class probabilities for $\mathbf{x}$ follow a uniform distribution. In such cases, the final predicted class $\hat{y} = \mathrm{argmax}\sigma(M(\mathbf{x}))$ lacks a dominant class probability, indicating high uncertainty and hinders confident predictions and effective learning. Therefore, it is important to shift the model's optimization objective toward samples with high uncertainty to improve its performance in these challenging regions. To that end, we propose *Balanced Uncertainty Sampling* (BUS) built upon the concept of Uncertainty Sampling (Lewis, 1995) but with label balancing awareness for avoiding performance degradation due to training biases (Johnson & Khoshgoftaar, 2019). Let $\bar{D}_2^{\eta} = \cup_{i=1}^{\eta-1} D_2^i \cup D_1$ be the union of accumulated subset from previous $\eta - 1$ iterations in Phase B and subset $D_1$ from Phase A. At each iteration $\eta$, we apply the score function in Eq. 5 to all sample $(\mathbf{x}_i, y_i) \in D_P \backslash \bar{D}_2^{\eta}$ and choose $\epsilon$ highest score samples to form $D_2^{\eta}$.

$$J_2(\mathbf{x}_i, y_i) = \alpha \exp\big(-H(\mathbf{x}_i)\big) + (1-\alpha)\big(1 - \exp(|\bar{D}_2^{\eta y_i}|/|\bar{D}_2^{\eta}| - 1)\big) \tag{5}$$

where each data sample in $D_2^{\eta}$ is selected via its prediction uncertainty (first term) and the imbalance ratio of its labels among all previously chosen samples so far (second term) and $\alpha \in [0, 1]$ is a regulation parameter to balance these terms (default $\alpha = 0.5$) for performance boosts.

**Active model updating.** At each iteration of Phase B, model $M_2$ is updated by training on the newly acquired sample set $D_2^{\eta}$. However, repeatedly focusing on new subsets risks forgetting earlier and less frequently selected samples (Goodfellow et al., 2014), thus lowering the overall performance as we observed in our preliminary experiments. To mitigate this, we expand $D_2^{\eta}$ to include similar samples from the same clusters identified in Phase A, forming an augmented set $D_{2u}^{\eta} = D_2^{\eta} + C(D_2^{\eta})$. The intuitions behind this choice are: (i) if the enquired samples $D_2^{\eta}$ might not be enough for the model to correct itself, then the samples in the same cluster $C(D_2^{\eta})$ can help to reinforce the result and (ii) we *replay* the memories (Rolnick et al., 2019) to avoid the forgetting phenomenon. Choosing the additional samples also implicitly incorporate diversity alongside informativeness in sampling suitable subsets for better performance.

## 2.3 PHASE C - CROSS DEVICE ADJUSTMENT

Phase C, *the most important and interesting part of ActSCA*, addresses the problem of effectively attacking the target device in the cross-device settings. Variations in signal waveforms among devices due to many factors such as manufacture variations or operating temperatures can significantly reduce the prediction accuracy of the DL models when attacking target devices due to input mismatches (Bhasin et al., 2020). The key idea of Phase C is to exploit the training pool $D_P$ to construct a new device-specific model $M_3$, which is specially tailored to match the target device characteristics by updating $M_2$ with a data collection $D_3 \in D_P$ containing most similar traces to $D_T$. This scheme helps to improve the performance without requiring any additional training data from the target device like existing works (Yu et al., 2023; 2021). We call this approach a *reverse* training scheme. It also contains two steps (cf. Algorithm 5 in Appendices for pseudo-codes).

**Similarity matching.** This step aims to find a set of traces $D_3 \in D_P$ that can reflex the characteristics of the victim traces $D_T$ so that our model can adapt to the target traces via two key aspects: *feature* and *prediction* viewpoints of $M_2$. Given two traces $\mathbf{x}_p \in T(D_P)$ and $\mathbf{x}_t \in T(D_T)$, we define two functions $d_P(\mathbf{x}_t, \mathbf{x}_p)$ and $d_F(\mathbf{x}_t, \mathbf{x}_p)$ for prediction and feature dissimilarity as follows:

$$d_P(\mathbf{x}_p, \mathbf{x}_t) = \sum \sigma(M_2(\mathbf{x}_p)) \log \frac{\sigma(M_2(\mathbf{x}_p))}{\sigma(M_2(\mathbf{x}_t))} \tag{6}$$

$$d_F(\mathbf{x}_p, \mathbf{x}_t) = ||\widetilde{M_2}(\mathbf{x}_p) - \widetilde{M_2}(\mathbf{x}_t)||_2^2 \tag{7}$$

Here $d_P$ measures dissimilarity between $\mathbf{x}_p$ and $\mathbf{x}_t$ from $M_2$ prediction outcomes via Kullback–Leibler (KL) divergence (Kullback & Leibler, 1951) and $d_F$ represents dissimilarity between $\mathbf{x}_p$ and $\mathbf{x}_t$ via their feature vectors obtained from the feature layer $\widetilde{M_2}$ of $M_2$ under $L_2$-norm. The general intuition is that if two traces $\mathbf{x}_p$ and $\mathbf{x}_t$ are similar, they should have close features and prediction outcomes from $M_2$. Let $\kappa$ be the number of expected similar samples for each $\mathbf{x}_t \in T(D_T)$ and $\beta \in [0, 1]$ be a regulation parameter to balance two dissimilarity viewpoints (default $\beta = 0.5$). For each trace $\mathbf{x}_t \in T(D_T)$, $D_3$ will contain $\kappa \cdot \beta$ and $\kappa \cdot (1 - \beta)$ most similar samples to $\mathbf{x}_t$ under the prediction and the feature views of $M_2$, respectively, i.e.,

$$D_3 = \left( \oplus_{\mathbf{x}_t \in D_T} KL(\kappa \cdot \beta, \mathbf{x_t}) \right) \bigoplus \left( \oplus_{\mathbf{x}_t \in D_T} KNN(\kappa \cdot (1 - \beta), \mathbf{x_t}) \right) \tag{8}$$

where $KL(m, \mathbf{x_t})$ and $KNN(m, \mathbf{x_t})$ be a collection of $m$ most similar sample $(\mathbf{x}_p, y_p) \in D_P$ of $\mathbf{x}_t \in T(D_T)$ under dissimilarity functions $d_P$ and $d_F$, respectively, and $\oplus$ is a concatenation/join operator between two collections.

**Target device specific model creation.** Having $D_3$, we now can construct $M_3$ to specifically attack the target device $TD$. Wlog, assuming that the model $M_2$ is a conventional deep learning model which includes a multi-layer backbone $\widetilde{M}$ followed by a fully connected layer $\sigma$ for prediction. We proposed to *freeze* the weight of $\widetilde{M}$ and only update the classification head $\sigma$ with training data $D_3$. The rationale behind this design is that after phase B, we already have a model $M_2$ that works well on the current device and able to extract valuable features. Hence, we do not need to further refine it to save computation cost. At the end, we have a new model $M_3$ which is specially tailored to the target device and can be used to effectively attack it.

## 3 EXPERIMENTS

We demonstrate performance of ActSCA on a PQC CRYSTAL-Kyber using power leakage to retrieve 300 different secret keys from multiple target devices under various SOTA DL-SCA architectures such as MLP (Martinasek et al. 2014), Transformer (Hajra et al., 2024) and particularly CNN (Hoang et al., 2024) (due to its abilities to mitigating effects of signal jittering (Cagli et al., 2017) and de-synchronisation (Emmanuel et al., 2018)). The number of attack tracks in $D_T$ varies from 10 to 100. Note that most existing works only report evaluations on a single key, which is lack of generation and can lead to biased results. Unless otherwise specified, we use CNN as the base model, 10 attack traces and default parameters of ActSCA. Details about experiment settings, model architectures and hyper-parameter turnings can be found in Section C in Appendices. We also illustrate the generability of ActSCA on other (protected) devices including 5 Kyber and 10 non-PQC AES (Emmanuel et al., 2018), another cryptographic algorithm (cf. Section G in Appendices).

**Evaluation metrics.** We use two different metrics: Mean-rank and Top-$k$ accuracy (or Success rate of order $k$) (Papagiannopoulos et al., 2023). Mean-rank calculates the average rank of the correct key over multiple runs on a device. Top-$k$ accuracy counts the number of times the studied key is ranked in the first $k$ keys over multiple runs. Results are averaged over 10 runs.

**Evaluation baselines.** For ActSCA, we specially study its Phases B and C (i.e., models $M_2$ and $M_3$ in Figure 2). Phase B demonstrates the ability of ActSCA to provide a more efficient DL model without using the full training pool $D_P$. Phase C shows the ability of ActSCA in cross-device attacks by exploiting the pool $D_P$. We compare our method ActSCA with Baseline, which is the employed DL model in ActSCA (e.g. CNN, MLP or Transformer) and trained with the full training pool $D_P$. We also compare with other relevant ML methods like Core-set (Sener & Savarese, 2017), JTT (Liu et al., 2021), DynUn (He et al., 2024) and InfoMax (Tan et al., 2025) for data selection of Phase A of ActSCA. For some other methods like Dual-Leak (Yu et al., 2023), while directly comparing to ActSCA is infeasible due to different algorithmic targets and huge amount of cross-device trace requests, we adapt relevant parts of their techniques to compare to Phase B of ActSCA like different data sampling strategies (Yu et al., 2023). For portability, we compare Phase C with various SOTA approaches such as MDMSD (Wu et al., 2023), ZMUV (Montminy et al., 2013), MMD (Cao et al., 2021) and ADA (Wang et al., 2023).

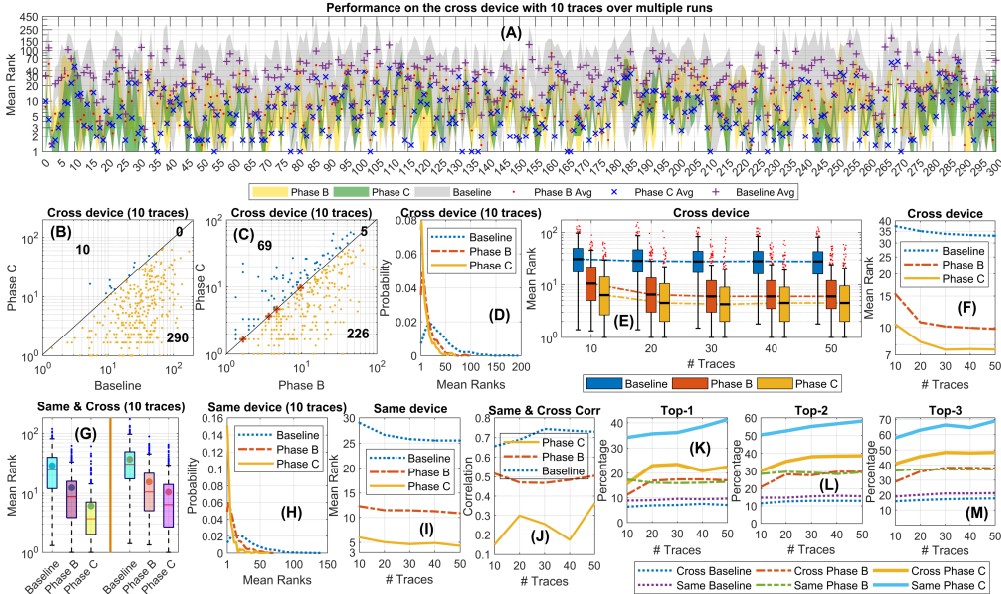

Figure 3: Performance of our method ActSCA on both the same (*profiling*) and cross (*attacking*) devices with 300 test keys and $\{10, 20, 30, 40, 50\}$ attack traces per key (default is 10). *Baseline* denotes the model trained on the full pool of 200K traces. *Phase B and C* are the two phases of ActSCA as presented in Section 2 above. **Overall**, our method ActSCA dramatically improves the performance compared to the baseline with up to 4.5x and 5.9x improvements on the averaged mean ranks of 300 test keys on the cross and same devices compared to the Baseline, respectively.

## 3.1 MAIN RESULTS

Figure 3 show performances of ActSCA with 300 test keys on the same (profiling) and cross (attacking) devices using the CNN architecture with 10 to 50 attack traces (default $D_T = 10$).

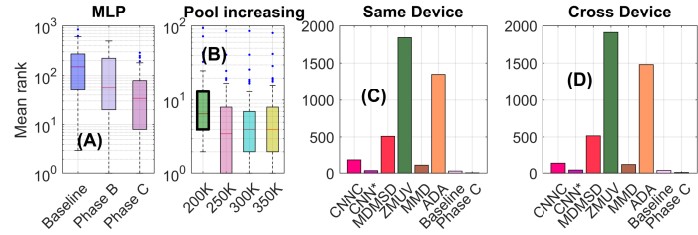

Figure 4: Other experiments on ActSCA.

**Mean-rank accuracy on the cross device.** Figure 3 (A) shows the $[min, max]$ and averaged mean-ranks of Baseline, Phase B, and Phase C. Phase C clearly acquires the best performance while Baseline is the worst one. (B) further compares Baseline and Phase C with a scatter plot, where each point below the diagonal line indicates a (much) better performance for Phase C on a specific test key. Over 300 keys, Phase C surpasses Baseline on 290 keys (i.e. 96.7%). Similarly, (C) shows that Phase C is better than Phase B on 226/300 keys (i.e. 75.3%). (D) shows the distributions of mean-rank values, where Phase C outperforms others with much higher probability to generate lower mean-rank values. (E) shows the mean-ranks of all 300 test keys with 10 to 50 attack traces, while (F) shows their averaged values. Phase C significantly outperforms others in terms of both averaged mean-ranks and stability of results (smaller box sizes). E.g., with 10 traces, the averaged mean ranks are 10.4, 15.4, and 37.3 for Phase C, Phase B and Baseline, resp. Moreover, the more attack traces we use, the better the performance improvement. Overall, on the cross device, Phase C dramatically improves the attack performances of 3.5x to 4.5x compared to Baseline. Even without target device adjustments, Phase B is already better than Baseline from 2.4x to 3.3x.

**Top-$k$ accuracy on the cross device.** Figure 3 (K, L, M) presents the top-$k$ performance of ActSCA and Baseline on the cross device. For 10 traces, Baseline has top-1 accuracy of 6.4%, while Phase B and C have 11.3% and 15.8%, resp. For 50 traces, Baseline slightly increases to 7.1%, while Phase B and C raise significantly to 17.1% and 22.5% (i.e. 2.4x and 3.2x better), resp. The same results are seen on top-2 and top-3. The more attack traces, the larger the gaps between ActSCA and Baseline.

**Performance on the same device.** Figure 3 (G, H, I) shows the results of ActSCA on the profiling devices. Similarly, ActSCA dramatically improve Baseline in both mean-ranks and top-$k$ scores.

**The same and cross device relationships.** Since the training data comes from the profiling device, its attacked performances are better than in the target device as seen in Figure 3 (G). Moreover, the performances of ActSCA on the profiling and target devices are highly correlated as seen in Figure 3 (G, K, L, M). Hence, we can consider the profiling device performances as (soft) upper bounds when attacking target devices. However, for each key, correlations between profiling and attack devices in Phases B and C are lower than Baseline due to their partial training data.

### 3.2 OTHER STUDIES

**Other DL architectures.** Figure 4 shows deeper performance studies of ActSCA on $TD$ with 10 attack traces. As a *generic framework*, ActSCA can be used to boost any base model. Figure 4 (A) shows that both Phases B and C help to dramatically improve the performance compared to Baseline using MLP and Transformer.

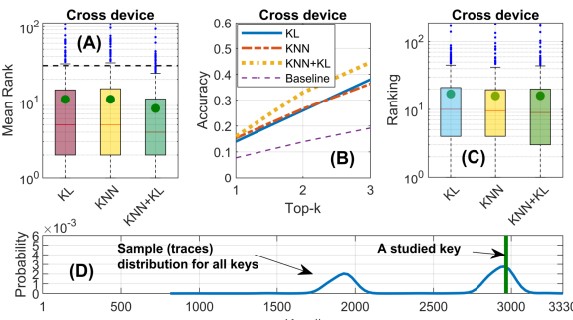

**Having more data?** After having $M_2$, we can *add new data to the pool $D_P$ without recreating models*, thus saving computation cost. When attacking target devices, the extended pool $D_P$ will be employed. Figure 4 (B) shows the performance of Phase C when $|D_P|$ is expanded by 50K to 150K samples. Larger $D_P$ lead to more diverse data and improved performance.

Figure 5: Performance of Phase C with different neighborhood selection scheme including: KL, KNN and KNN+KL on the cross device. Dashed line presents Baseline performances, circles denote averaged values.

**Comparisons to other approaches.** Figure 4 (C, D) compares ActSCA (Phase C), our CNN Baseline, CNN and CNNC (Hoang et al., 2024) and portability methods including MDMSD (Wu et al., 2023), ZMUV (Montminy et al., 2013), MMD (Cao et al., 2021) and ADA (Wang et al., 2023). ActSCA acquires much better performance (from 5.5x to 367.8x on the same and from 3.8x to 203.4x on the cross devices) compared to all others in terms of mean-ranks.

### 3.3 ABLATION STUDIES

We study in depth the characteristics of all Phases of ActSCA (cf. Appendix D for parameter effects and other analyses).

**Neighborhood strategies of Phase C.** Figure 5 (A) shows averaged mean-ranks of Phase C with different similarity matching schemes (c.f. Section 2.3). KNN+KL with both feature and prediction views acquires the best performance with 8.24 compared to 10.88 and 10.89 of KL and KNN, resp. KNN+KL also dominates others in top-$k$ scores (B). In (C), we sort all test keys $k$ based on their total appearances in $D_3$ and report their ranks. KNN+KL has the lowest averaged ranking of 15.8 compared to 16.0 and 16.9 of KNN and KL, respectively.

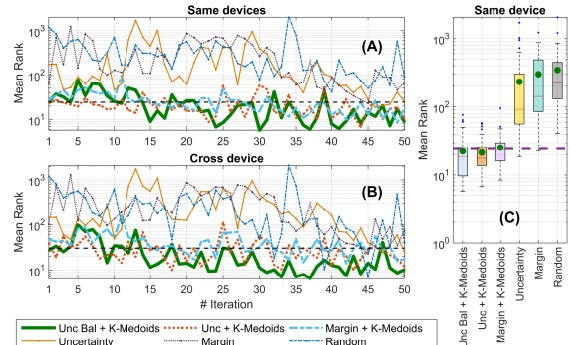

Figure 6: The performances of different active selection strategies of Phase B for both the same (**A**) and cross (**B**) devices on each iteration. (**C**) Whisker box-plot of different strategies on all iterations (blue circles denote averaged mean ranks). The dashed horizontal lines show the performance of Baseline.

It means that KNN+KL puts more traces with correct keys into $D_3$ to build $M_3$, thus creating implicit biases toward correct keys and possibly improving performance. An example is shown in (D), where key $k$ of the cross device has the highest number of traces in $D_3$.

**Active selection strategies of Phase B.** Figure 6 shows performances of Phase B on each iteration compared to 6 selection strategies including base methods (Uncertainty, Margin and Random sampling) (Settles, 2009; Yu et al., 2023) and our clustering-based methods (Uncertainty Balance,

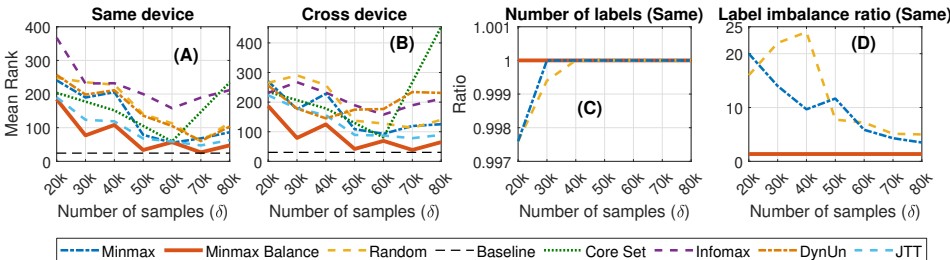

Figure 7: Performance of different selection strategies of Phase A for the same and cross device.

Uncertainty and Margin with $k$-medoids). For both devices, Random strategy performs the worst. And Unc Bal + $k$-medoids acquires the best overall performance and become stable when having enough iterations, e.g., outperforms Baseline on 39/50 iteration (i.e. 78%) on the cross device.

**Selection strategies of Phase A.** Figure 7 (A, B) shows averaged mean ranks for different strategies wrt. $|S|$. Maxmin Balance is significantly better than others while Random performs the worst. When $S$ is larger, the performance comes closer to Baseline. In a few cases (e.g. 50K and 70K), Maxmin Balance acquires almost the same performance with Baseline with 200K training traces. (C, D) show the total numbers of labels and imbalance ratios among labels in $S$. When Maxmin Balance has full labels for all cases, while others only have when $\delta$ is large enough. Moreover, while imbalance ratios of Random and Maxmin are very high, Maxmin Balance has very stable ratios, ranging very slightly around 1.36 due to its special label balancing strategy (cf. Section 2).

## 4 RELATED WORK AND DISCUSSION

**Deep Learning based SCA (DL-SCA).** As mentioned in Section 1, DL has become one of the most effective approaches for profiling SCA attacks with many successful reports, e.g., (Maghrebi et al., 2016; Aydin et al., 2020). The employed DL architectures, side channels, and targeted cryptography schemes are also diverse. But CNN and MLP are the two most commonly used (Martinasek & Malina, 2014; Emmanuel et al., 2018) besides a few others like Transformer (Hajra et al., 2024) and LSTM (Cagli et al., 2017). E.g., CNN and power tracks are used to attack PQC NewHope and FrodoKem (Kashyap et al., 2021) and CRYSTAL-Kyber (Hoang et al., 2024). All these techniques relies on large amount of data to train their ML models. However, none of them considers the fact that increasing data size may not be as effective as using smaller sets of selected data to improve the performance like ActSCA. A few like (Wouters et al., 2020; Rijsdijk et al., 2021) aims to find smaller networks to reduce computation time, which fundamentally differs to ActSCA.

**Portability problem.** Portability has recently emerged in SCA research following the needs for more practical attacking scenarios (Bhasin et al., 2020). All existing works tackles the problem via either data or model perspective but not both like ActSCA. E.g., (Montminy et al., 2013) normalizes the means and variances of target traces to have the same statistical properties with profiling traces. Some like (Das et al., 2019; Rioja et al., 2020) use multiple devices to enrich training data for reducing overfitting. However, they do not create explicit models for cross devices specifically like ActSCA. ActSCA also reduces overfitting in a reverse way, reducing its training data. MDMSD (Wu et al., 2023) removes layers of pretrained models to have portability. Dual-Leak (Yu et al., 2023) applies Deep Supervised Active Learning to select traces from a pool of unlabeled traces of the target device. Others employ transfer learning (Yu et al., 2021; Cao et al., 2021; Wang et al., 2023) to reduce required traces on the target device. However, all of them still consume large amount of target traces, which may be infeasible in reality. ActSCA, due to its unique reverse training scheme on the profiling pool, only needs to collect one or a few target traces and thus is much practical.

## 5 CONCLUSION

We introduce ActSCA, a *unique* framework which can be used to boost the performance of any base DL models in SCA. ActSCA particular aims at two key problems: (i) how to improve performance of a DL-SCA model on a training data pool and (ii) how to effectively cope with the device portability problem. We evaluate the performance of ActSCA on PQC Crystals-Kyber and AES cryptography with multiple devices as case studies. Extensive experiments demonstrate that ActSCA help to dramatically boost performance of employed base models and can successfully retrieve secret keys using only a few unlabeled attack traces from target devices, thus making it a very practical method.

## 6 REPRODUCIBILITY STATEMENT

The AES datasets used in this study are publicly available, with sources cited in the dataset descriptions. Our source code is provided in the supplementary material. Additionally, the Kyber dataset is available from the authors upon request due to its large size.

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

# Appendices

As a generic framework, ActSCA can be applied for any side channel profiling attacks. In this paper, we evaluate the performance of our ActSCA framework on attacking two cryptography schemes including PQC CRYSTAL-Kyber and non-PQC AES on different hardware/software implementations and protected schemes (5 devices for Kyber and 10 devices for AES) using power leakage. Due to space constraints, besides the main results on CRYTAL-Kyber presented in the main paper, the remaining results of all other CRYSTAL-Kyber and AES devices, deep analyses on characteristics of ActSCA and various algorithm design choices, theoretical proof for Theorem 1 on Phase A of ActSCA, and experimental settings can be found below.

**Overviews.** Our appendices are outlined as follows. In Section A, we introduce our attacking threat model. We present the background of PQC CRYSTAL-Kyber and non-PQC AES cryptography in Section B. Then, we describe the data collection setting and our DL model architecture with its training environment in Section C. In Section D, we discuss the behavior of our algorithm effected by different hyperparameters including parameter settings for Phases A, B and C of ActSCA. In Section E, we present more interesting characteristics of ActSCA including: (i) performances on different architectures such as Transformer, MLP and CNN; (ii) runtime and memory consumptions; (iii) adding more traces to the pool; (iv) performance on Guessing Entropy; (v) the choice of clustering method in Phase A; (vi) why do we need to partly freeze models in Phase C; (vii) attacks with higher numbers of traces; and (viii) numbers of attack evaluations. Further attacking results on 5 other CRYSTAL-Kyber devices can be found in Sections F. In Section G, we analyse performances of ActSCA on AES crytography including: (i) performances on 5 unprotected AES devices; (ii) performances on the ASCAD protected device; (iii) robustness of ActSCA on the choice of distance functions under AES protecting schemes; (iv) performances of ActSCA with different base models; and (v) performances of ActSCA with other 4 protected devices from the AES_PTv2 datasets. We also provide additional related works in Section H. Detailed and High-level pseudo-codes of our algorithm and theoretical proof for Theorem 1 for Phase A can be found in Section I. Broader impact is shown in Section J. Limitations of our method is discussed in Section K. LLM declarations can be found in Section L.

**Summary.** Overall, ActSCA significantly outperforms all SOTA methods for portability on both CRYSTAL-Kyber and AES under different hardware, software, and protection settings. It also can significantly boost performances of all employed base models (baselines) including MLP, CNN, Transformer, InceptionNet, CNN_best, and EFCNN. The stronger the base models, the stronger the performance of ActSCA. It is also quite robust to the choices of hyperparameters, clustering methods and distance functions. Moreover, the more attack tracks, the better the performance of ActSCA. And the more we expand the data pool $D_P$, the better the performance of ActSCA during the attacking phase (without retraining the whole model).

CONTENTS

## A  THREAT MODEL

In this paper, our threat model is built upon the common threat model of the SCA profiling attack described in Sections 1 and 2 with a key difference on the trace limitation in the attacking phase. Concretely, let assume that there is a *profiling device* and an *attacking device* where both of them exploit Kyber, AES or any other encryption scheme for information protection purpose.

- Attacker's goal: The goal of the attacker is to retrieve accurately secret key of the *attacking device* under the limited number of power traces.
- Attacker's capability: The attacker has an access to any specific functions of Kyber or AES algorithm and control CWLite to capture energy consuming by these functions on both devices. In addition, the attacker is able to measure as many traces as possible on the *profiling device*. However, the key difference to the common threat model (Wu et al., 2023; Cao et al., 2021) is that we only have a very small set of traces on the *attacking device*. This make our attacking model a highly practical one since one needs to have the target device at hand to measure side channel data, which is not always feasible. Moreover, there is no limitation on computational resource for the attacker.

## B  BACKGROUND ON CRYSTAL-KYBER

**PQC CRYSTALS-Kyber.** PQC CRYSTALS-Kyber (Bos et al., 2017) is a *public-key encryption* scheme that consists of three algorithms: Key generation ($\mathcal{K}$), Encryption ($\mathcal{E}$), and Decryption ($\mathcal{D}$). Generally, the algorithm performs key generation to generate a public key ($pk$) and secret key ($sk$). Any message ($m$) will be encrypted to ciphertext ($c$) by the encryption algorithm ($c = \mathcal{E}(m, pk)$) to hide information under an incomprehensible form. Then, this text can be securely transported over a communication channel before being decoded by decryption algorithm to recover the original

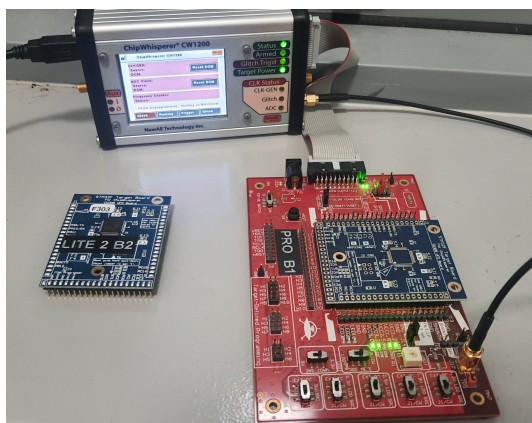

Figure 8: The profiling (left) and attacking (right) devices together with Chipwhisperer platform used in the paper.

text ($m' = \mathcal{D}(c, sk)$). The core algorithm of Kyber comes from the first LWE-based encryption scheme (Regev, 2009) with the replacement of $\mathbb{Z}_q$ ring by polynomial ring and constraints that both the secret and error vectors have small coefficients.

**Input:** Secret key $sk \in \mathcal{B}^{12.k.n/8}$, Ciphertext $c \in \mathcal{B}^{d_u.k.n/8+d_v.n/8}$
**Output:** Message $m \in \mathcal{B}^{32}$
  1: $\mathbf{u} := \text{Decompress}_q(\text{Decode}_{d_u}(c), d_u)$
  2: $v := \text{Decompress}_q(\text{Decode}_{d_v}(c + d_v.k.n/8), d_v)$
  3: $\hat{\mathbf{s}} := \text{Decode}_{12}(sk)$
  4: $m := \text{Encode}_1(\text{Compress}_q(v - \text{NTT}^{-1}(\hat{\mathbf{s}} \circ \text{NTT}(\mathbf{u})), 1))$
  5: **return** $m$

**Algorithm 1:** KYBER.CPAPKE.Dec($sk, c$): decryption

**Side Channel Analysis targeting CRYSTALS-Kyber.** Being known to resist quantum computer attacks based on introducing randomness to data during encrypting and decrypting, CRYSTALS-Kyber still faces the vulnerability of side channel attacks due to the ineffective implementation on popular physical architectures such as ARM Cortex-M4. For example, in the implementation provided by the PQM4 library (cf. Algorithm 1 in Appendices), the secret key $\hat{s}$ and the ciphertext $\hat{u} = NTT(u)$ are partitioned into four segments $\{s_0, s_1, s_2, s_3\}$ and $\{u_0, u_1, u_2, u_3\}$ before conducting double base polynomial multiplication. Each $s_i$ is considered as a coefficient with the value as an integer number in the range $[0 \ldots 3328]$ for the Kyber512 scheme. After that, the algorithm uniquely selects one coefficient $s_i$ and multiplies this coefficient by a part of the ciphertext to recover the original information. Therefore, each coefficient is used for a period of time and consumes a segment of power trace, which can be exploited to reveal its value in previous works such as (Hoang et al., 2024).

## C EXPERIMENT SETTINGS

**Devices and data collections.** The data collection process is conducted following a CRYSTAL-Kyber attack model described in (Hoang et al., 2024). Figure 8 illustrates a synchronous system in our study with a ChipWhisperer Lite (CW Lite) (O'Flynn & Chen, 2014) with CW308 UFO baseboard and a CW308T-STM32F3 Cortex-M4 microcontroller. Concretely, we implement Kyber512 decryption algorithm of PQM4 libarary (Kannwischer et al., 2019) on the STM32F3 microcontroller. Our ciphertexts are generated by CPAPKE.Enc to randomly form 32-byte plaintext. Then, we perform a power consumption measurement by triggering CW Lite right after the decryption phase activation to attack the *doublebasemul* function in Kyber assembly code as described in (Hoang et al., 2024). The CW Lite will capture signals with a sampling rate of $4 \times 7.37$ MHz and 10 bits ADC 105 MS/s resolution. Finally, we gather the dataset with information including: secret coeffi-

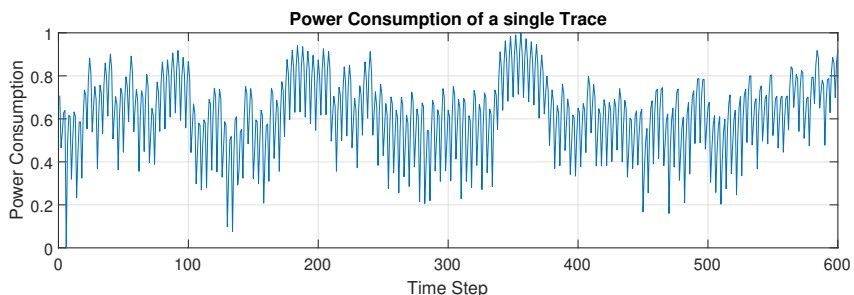

Figure 9: An example of a measured trace of CRYSTALS-Kyber.

cient keys, ciphertexts and power traces. *Concretely, the coefficient keys are integers from [0-3328] and will be targets for the attacks while power traces (600 points each) are inputs for base Deep Learning models.* Figure 9 shows a collected power trace as an example.

**Training, validation and testing.** For training DL models, we collect a training data pool $D_P$ of 200K samples. A validation set of 50K traces on the profiling device is also created to evaluate the training process. For testing DL models, we randomly choose 300 arbitrary keys and generate for each key 100 traces on the profiling device and 100 traces on the attacking device. These keys and their corresponding traces will be used to evaluate the attacking performance on the profiling (*same*) and attacking (*cross*) devices.

**Model architecture and hyper-parameters.** We employ various common ML architectures in hardware security (Martinasek & Malina, 2014; Cagli et al., 2017; Emmanuel et al., 2018) to evaluate the performance of our algorithm ActSCA including: CNN, MLP and Transformer. These models are trained with Cross Entropy loss and RMSprop is chosen as the optimiser with $epsilon = $ 1e-7 and $momentum = 0.0$. The learning rate is set at 1e-6.

The studied CNN model consists of 5 Convolutional Layers to extract the time series features, followed by 5 layers of Dense Layers and a Softmax Layers at the end to make the prediction. Similarly, the MLP model consists of 5 Dense Layers for feature extraction, followed by 5 layers of Dense Layers and a Softmax Layer at the end. The output and input dimension of each layer in MLP and CNN are the same, the main difference is we use convolutional with kernel size of 3 for CNN while linear layer are used in MLP. In the Transformer architecture, we first use 4 CNN layers to extract features, followed by 6 self-attention layers. Each attention layer has 8 heads with the dimension of the output head of 32.

**Implementation details.** All experiments were conducted on a workstation equipped with an NVIDIA A5000 GPU (24GB VRAM), an AMD Ryzen 9 5950X CPU, and 128GB RAM, running Ubuntu 20.04. The models were implemented using Tensorflow and trained with CUDA 11.7 support. For clustering (e.g., K-Medoids), we used the scikit-learn-extra library, and other preprocessing routines were implemented using NumPy and scikit-learn. We fixed the random seed to ensure reproducibility. All experiments were executed on a single GPU unless otherwise stated. Training scripts and evaluation pipelines were managed using custom Python code and executed in a controlled environment to minimize performance variance.

**Evaluation Metrics.** We use two main metrics: Mean-rank (Emmanuel et al., 2018) and Top-$k$ accuracy (or success rate at order $k$). Mean-rank calculates the average rank of the correct key over multiple runs on a device. Top-$k$ accuracy counts the number of times the studied key is ranked in the first $k$ keys over multiple runs. Since Mean-rank provides an overall average, it may overlook individual key performance, whereas Top-$k$ accuracy offers an additional view by assessing each key independently. We also report Guessing entropy (Wu et al., 2020), which reflects the expected number of attempts an attacker would need to identify the correct key, assuming they test key candidates in order of decreasing likelihood as determined by the attack.

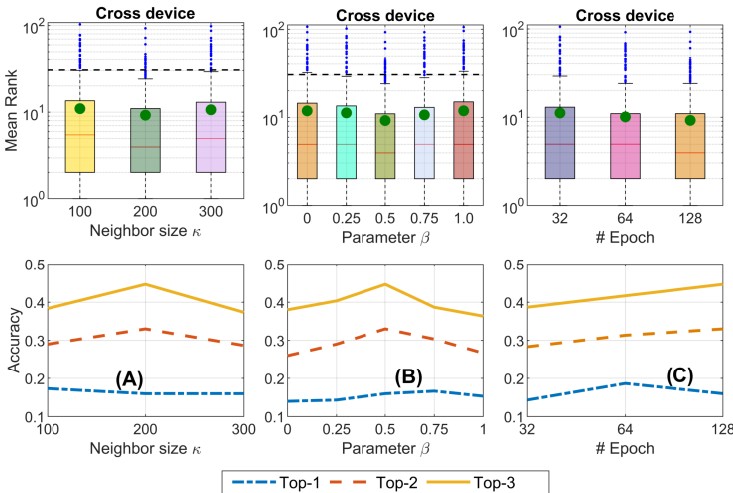

Figure 10: Effects of different parameters on Phase C of ActSCA on the cross device. Baseline is shown as dashed horizontal lines. Circles denote averaged values.

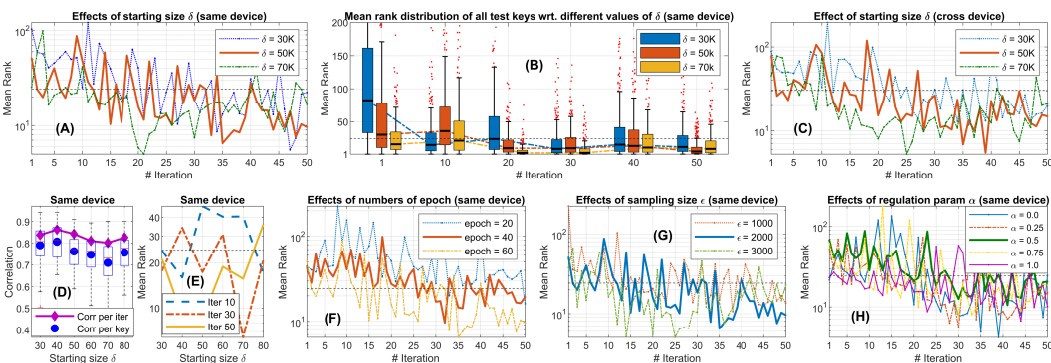

Figure 11: Effects of different parameters on Phase B of ActSCA for both the same and cross devices. The dashed horizontal line shows the averaged mean ranks of Baseline for the same and cross devices.

## D ABLATION STUDIES ON ACTSCA (CONT.)

**Effects of parameters on Phase C.** Phase C of ActSCA has 3 main parameters including: the neighborhood size $\kappa$, the regulation parameter $\beta$ and the number of update epochs. Their effects on ActSCA are studied in Figure 10. When $\kappa$ increases, the performance increases slightly since we have more data for creating the ML model $M_3$. However, too many data can lower important traces related to the correct keys, thus reducing accuracy (A). Hence, we use $\kappa = 200$ as a default value. The param $\beta$ is used to balance the neighborhood queries towards data or model viewpoints and is fixed as 0.5, which is also the best value as shown in (B). In (C), to few update epoch will make the model not converge well. However, too many epoch will lead to overfitting problem. Both reduce the overall performance. Hence, we fix the update epoch as 128.

**Active selection strategies of Phase B (cont.).** How many iteration is enough? The correlations between the same and cross devices for clustering-based methods are very high (0.73 to 0.89) for the selected method Unc Bal + K-Medoids (the same for all others clustering-based strategies). Hence, during the training phase, we can use the profiling device to find the likely best model for the attacking device using the validation set. E.g., at iteration 36, we have the best model for the profiling device and it is also the best model for the cross device as shown in Figure 6 (A) and (B).

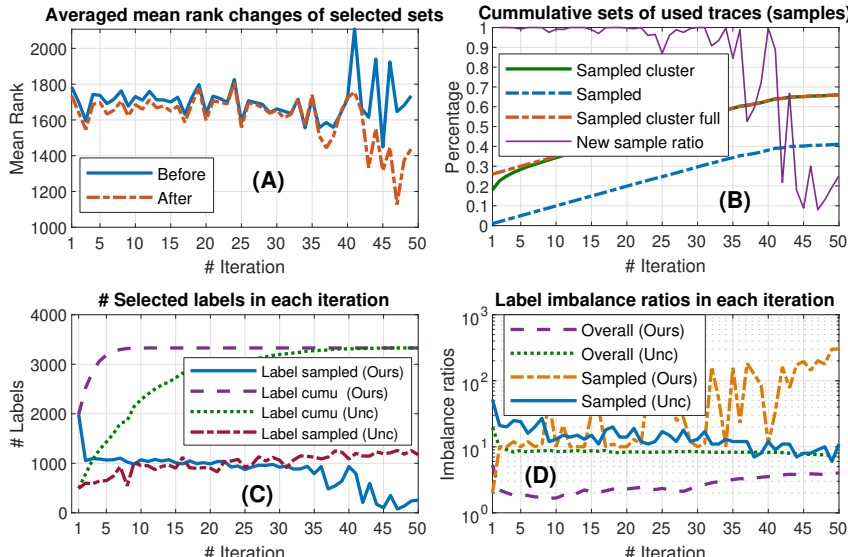

Figure 12: Characteristics of Phase B.

**Effects of parameters on Phase B.** Phase B of ActSCA has a few parameters including: the starting size $\delta$ for the training data $D_2$, the sampling size $\epsilon$ for each iteration, the regulation param $\alpha$ for balancing training sets, and the number of update epoch in each iteration. Effects of these parameters are studied in Figure 11.

The effects of $\delta$ are shown in (A-C), in which the averaged mean ranks show clear decrease trends wrt. each iteration and consistently beat baseline when $\delta$ is large enough ($\delta > 15$). When $\delta$ is larger, mean rank decreases to lowest peaks faster before moving up again. This matches with our data redundancy problem discussed in Sections 1 and 2 that increasing training data may not be a good solution for performance gains. The best performance is usually acquired when $\delta = 40K - 70K$. Hence, we choose $\delta = 50K$ as a default param. In (D), we measure the correlations of the performance during all 50 iterations of Phase B wrt. different $\delta$ (i.e. corr per iter). They ranges from 0.83 to 0.86, indicating the stability of our active updating scheme of Phase B. Similarly, we measure correlations on the mean ranks of each key between two consecutive iterations and shows the results using Whisker boxplot with blue circles denotes the averaged values wrt. different $\delta$. The high correlation here indicates that the mean rank of each key decreases stably at each iteration.

Effects of number of updating epoch for each iteration is show in (F). When $epoch$ increases from 20 to 60, the performance of Phase B increases significantly. However, when $epoch$ is too large, overfitting can happen. Thus, we use 60 epoch as the default param. (G) shows effects of the sampling size $\epsilon$ on the performance of ActSCA. Similar to $\delta$, the larger $\epsilon$, the faster Phase B reaches its peak performance before raising up again due to the data redundancy problem. (H) illustrates effects of the parameter $\alpha$ on the overall performance of Phase B. Here, $\alpha = [0.25, 0.75]$ create more stable results in the long term. Hence, we fix $\alpha = 0.5$. Similar to $\delta$, the mean rank performances on the same and cross devices are highly correlated together wrt. different values of $epoch$, $\epsilon$ and $\alpha$. E.g, correlations between the same and cross devices are 0.93, 0.81, 0.92 for 20, 40, and 60 epochs, respectively.

**How does Phase B work?** Figure 12 deeply analyses behaviors of ActSCA. Particularly, (A) shows the averaged mean ranks of each sample (trace) in the selected set $D_2^\eta$ ($\epsilon = 2000$) at each iteration $i$ (c.f. Algorithm 4) before and after the model update process. The averaged mean rank consistently improves on 91.8% cases. And the improvement is much larger at later iterations when the employed ML model becomes more stable. These proves the effectiveness of our model updating process in Phase B. In (B), we show the ratios of accumulated sets of samples over all the training pool from the selected set $D_2^\eta$, clustering extended set $D_{2u}^\eta = D_2^\eta + C(D_2^\eta)$ and the set of full used traces at each iteration of Phase B. At the beginning, most of samples in $D_2^\eta$ and $D_{2u}^\eta$ are new ones. ActSCA aims to explore unknown samples since they are more uncertain. However, the ratio of new samples

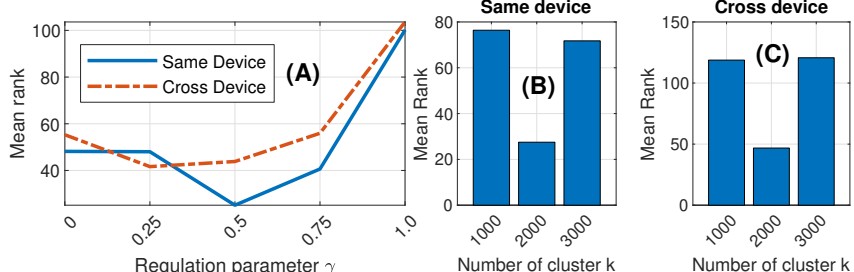

Figure 13: Performance of different trace selection strategies of Phase A for the same and cross device.

in $D_2^\eta$ reduces very quickly in last iterations. ActSCA tends to focus on seen samples that it cannot classify well. E.g., only around $10\%$ new samples are selected in iteration 45. At the end, Phase B only uses $66.1\%$ of the full 200K pool $D_P$ for training but having much better performance (cf. Figure 11).

We further look at another important point of Phase B: the label distribution problem. In (C), the number of labels (keys) selected at each iteration decreases over time on our method while increasing on Uncertainty sampling (US) (Yu et al., 2023), i.e., ActSCA aims to refine a smaller subsets of labels than US methods. By this way, it can focus on hard samples to better clarify it. However, sets of labels are more different at each iteration due to its special label balancing scheme (c.f. Section 2), leading to much faster cumulative rates compared to US. Since all labels are seen earlier, ActSCA has more time to adjust the model, thus leading to better performance. In (D), the imbalance ratios between the label with max sample and label with min sample are shown. For each sampled batch, the imbalance ratio of ActSCA is much higher than US, especially in late iterations since it also prioritizes the label balancing on all data it has seen so far to ensure that the model is not biased toward a specific set of labels. Hence, its overall imbalance ratio is much lower than US as seen in (D). This scheme leads to significantly performance improvements as shown in Figure 6.

**Effects of parameters on Phase A.** There are two key parameters in Phase A including: the regulation parameter $\gamma$ to control the label balancing in $S$ and the number of clusters $k$ for k-medoids. As shown in Figure 13 (A), the best values for $\gamma$ are around the median 0.5 for both devices. The number of clusters $k$ shows its best value at 2000 as shown in (B, C).

# E  FURTHER CHARACTERISTICS OF ACTSCA

Table 1: Mean rank of different model architecture.

| CNN | Transformer | MLP |
|-----|-------------|-----|
| 23.47 | 1619.6 | 165.11 |

**Different model architecture.** Table 1 shows the mean rank of various methods including multiple model architecture such as CNN, MLP, and Transformer. Transformer provided the worst performance compared to MLP and CNN. MLP and CNN performs $\sim 9.6\text{x}$ and $\sim 62.5\text{x}$ better than Transformer, respectively. This degradation in performance may stem from the mismatch between the Transformer's design and the nature of side-channel traces. Specifically, side-channel data typically exhibit strong local temporal dependencies—short-range patterns that convolutional networks are particularly effective at capturing due to their localized receptive fields. In contrast, Transformers are designed to capture long-range dependencies via self-attention mechanisms, which may introduce unnecessary complexity or even amplify noise in high-resolution side-channel signals, ultimately hindering performance.

**Computational complexity of ActSCA.** Table 2 shows memory and runtime of ActSCA and CNNC (Hoang et al., 2024). ActSCA is much more efficient due to its smaller data and model size.

Table 2: Runtime and memory size of ActSCA compared with CNNC.

| Method | Memory | Runtime |
|--------|--------|-----------|
| CNNC   | 14M    | 84s/epoch |
| PhaseB | 4M     | 40s/epoch |
| PhaseC | 4M     | 29s/epoch |

Table 3: Averaged mean ranks of ActSCA with 50 to 100 target traces.

| NumTrace | 50    | 60    | 70    | 80    | 90    | 100   |
|----------|-------|-------|-------|-------|-------|-------|
| Baseline | 24.91 | 23.12 | 22.01 | 22    | 22.14 | 21.47 |
| PhaseB   | 6.12  | 5.98  | 5.72  | 5.46  | 5.52  | 5.3   |
| PhaseC   | 3.52  | 2.62  | 2.4   | 2.4   | 2.38  | 2.24  |

**More attack traces.** Table 3 demonstrates mean-ranks of ActSCA with more attack traces. The more traces the better performances of ActSCA. E.g. Phase C significantly improves mean-ranks of 9.58x compared to Baseline, i.e., the chance of successful attacks increases dramatically by approximately 10x.

Table 4: Guessing Entropy of ActSCA (Baseline, Phase B and Phase C) compared to other baselines.

| NumTrace | MDSMD  | CNNC   | Transformer | MLP    | Baseline | Phase B | Phase C |
|----------|--------|--------|-------------|--------|----------|---------|---------|
| 50       | 515.75 | 206.24 | 1615.5      | 167.42 | 25.83    | 6.63    | 4.11    |

**Guessing-entropy.** In the context of side-channel analysis, Guessing Entropy (GE) is a metric that quantifies the average position (or rank) of the correct key among all possible key candidates after performing an attack (Wu et al., 2020). It reflects the expected number of attempts an attacker would need to identify the correct key, assuming they test key candidates in order of decreasing likelihood as determined by the attack. Table 4 shows the guessing-entropy of ActSCA vs. others SOTA methods such as MDSMD or CNNC using 50 traces. ActSCA boosts the results significantly (from 6x-125x). Results are consistent with those of mean-ranks.

Table 5: $k$-medoids vs $k$-means with different sample sizes in Phase A.

| #Sample   | 20K    | 30K   | 40K    | 50K   |
|-----------|--------|-------|--------|-------|
| k-medoids | 184.54 | 77.16 | 108.17 | 33.7  |
| k-means   | 393.53 | 85.27 | 74.84  | 31.64 |

**Phase A Cluster Flexibility.** In Phase A of ActSCA, a key idea is to find a small set of representative points for the whole data space, i.e. the point that is most similar to all other points inside its local group, which can be identified with any clustering methods. We choose to use $k$-medoids since each cluster centroid is a data sample. For other algorithms like $k$-means, since centers of clusters are not samples in the data, we can find a point that is closest to its center in each cluster as a representative. In Table 5 below, we study the effects of $k$-means and $k$-medoid on Phase A of ActSCA wrt. different initial samples on CRYSTAL-Kyber. These performances are comparable between clustering approaches, implies the clustering generalization of ActSCA.

Table 6: Updating strategy for Phase C.

| Method    | Baseline | Freeze | Unfreeze |
|-----------|----------|--------|----------|
| Mean rank | 13.2     | 8.7    | 889.7    |

**Phase C model freezing.** To support our design, we conduct an ablation study comparing 3 strategies: (i) a Baseline CNN trained directly on the target device, (ii) our proposed method where the model from Phase B ($M_2$) is frozen and only the classification head is updated using data from the target device (Freeze) (cf. Section 2.3 for details), and (iii) a variant where the entire model is unfrozen and fine-tuned on the target device (Unfreeze). As shown in Table 6, Freeze achieves a

significantly better mean rank (8.7) compared to the Baseline (13.2), demonstrating that M2 already provides useful representations for the target. In contrast, Unfreeze performs very poorly with a mean rank of 889.74, likely due to changes in the model learned representation. This empirically supports our decision to freeze $M_2$ in Phase C of ActSCA (cf. Section 2.3).

Table 7: Medians of Mean ranks of ActSCA and other SOTA methods over 10 and 100 runs and from 10 to 50 attack traces on the cross device setting. Best results are highlighted in bold.

| #Run | #Trace | 10 | 20 | 30 | 40 | 50 |
|------|--------|------|------|------|------|------|
| 10 runs | ActSCA | **10.04** | **9.39** | **8.7** | **8.48** | **8.36** |
| | Baseline | 37.33 | 35.03 | 33.78 | 33.26 | 32.96 |
| | MMD | 115.94 | 108.82 | 110.21 | 99.38 | 101.05 |
| | ADA | 1481.24 | 1506.49 | 1508.51 | 1496.39 | 1467.1 |
| | ZMUV | 1912.43 | 1875.06 | 1939.7 | 1932.63 | 1938.69 |
| | MDMSD | 515.41 | 542.68 | 513.39 | 485.11 | 468.95 |
| 100 runs | ActSCA | **9.92** | **9.73** | **8.53** | **8.26** | **8.07** |
| | Baseline | 35.73 | 36.04 | 33.31 | 34.8 | 34.42 |
| | MMD | 124.7 | 111.51 | 118.32 | 107.59 | 108.88 |
| | ADA | 1475.18 | 1412.56 | 1431.75 | 1579.21 | 1445.89 |
| | ZMUV | 1856.88 | 1930.61 | 2018.48 | 2003.33 | 1930.61 |
| | MDMSD | 481.07 | 501.27 | 538.64 | 470.97 | 499.25 |

Table 8: Medians of Mean ranks of ActSCA and other SOTA methods over 10 and 100 runs and from 10 to 50 attack traces on the same device setting. Best results are highlighted in bold.

| #Run | #Trace | 10 | 20 | 30 | 40 | 50 |
|------|--------|------|------|------|------|------|
| 10 Runs | ActSCA | **6.01** | **5.04** | **4.65** | **4.88** | **4.31** |
| | Baseline | 29.03 | 26.63 | 25.79 | 25.52 | 25.51 |
| | MMD | 107.44 | 114.51 | 116.53 | 111.48 | 100.37 |
| | ADA | 1348.74 | 1440.65 | 1511.35 | 1430.55 | 1428.53 |
| | ZMUV | 1842.74 | 1863.95 | 1840.72 | 1853.85 | 1881.12 |
| | MDMSD | 510.23 | 494.07 | 495.08 | 489.02 | 477.91 |
| 100 runs | ActSCA | **6.32** | **6.2** | **5.33** | **4.43** | **4.78** |
| | Baseline | 29.31 | 30.19 | 28.65 | 28.54 | 27.44 |
| | MMD | 112.61 | 108.57 | 124.73 | 131.8 | 117.66 |
| | ADA | 1489.23 | 1452.87 | 1442.77 | 1531.65 | 1523.57 |
| | ZMUV | 1932.47 | 1967.82 | 2021.35 | 1990.04 | 1891.06 |
| | MDMSD | 512.63 | 499.5 | 465.16 | 493.44 | 464.15 |

**Number of runs.** When developing our method, we initially evaluate the performance of ActSCA 50 times. However, the performance does not change much compared to running 10 times (as we demonstrate in Tables 7 and 8). Moreover, different from other works which mostly use 1 key, we evaluate 300 keys on Kyber, if we run experiments 50 times per key, we will need to run 300 x 50 = 15,000 times for 1 experiment, a massive amount of experiments. Hence, we decided to go with 10 runs, which also means 300 x 10 = 3,000 times per experiment (still a huge number) while having similar results.

Tables 7 and 8 show the medians of mean ranks of ActSCA and other SOTA methods on the cross and same device settings on Kyber using multiple attack traces with 10 to 100 runs, respectively. As we see, the performance does not change much between 10 and 100 runs for all methods. Moreover, ActSCA still outperforms others significantly (up to 379.1x).

Table 9: Mean ranks of ActSCA using additional data on CRYSTAL-Kyber.

| Method | Baseline | 0K | 5K | 10K | 50K |
|--------|----------|------|------|------|------|
| Mean rank | 13.2 | 8.7 | 8.82 | 8.18 | 7.12 |

**Additional data into the pool $D_P$.** Table 9 illustrates performances of ActSCA when the data pool $D_P$ is extended from 5K to 50K more samples (smaller than 50K-100K in Figure 4). ActSCA maintains strong attack performance even with no additional data (0K), ActSCA already achieves a mean rank of 8.7, which significantly outperforms the Baseline CNN's mean rank of 13.2 trained on the same 200K data. This highlights the effectiveness of our active learning and selection strategy even without expanding the dataset. When adding 5K or 10K more samples, ActSCA maintains consistent performance improvements (mean ranks of 8.82 and 8.18, respectively), and the trend continues with 50K additional samples, where the mean rank further improves to 7.12.

# F    PERFORMANCES OF ACTSCA ON MULTIPLE CRYSTAL-KYBER DEVICES

Table 10: Medians of mean ranks of ActSCA on other CRYSTAL-Kyber devices.

|         | Same  | Device 1 | Device 2 | Device 3 | Device 4 |
|---------|-------|----------|----------|----------|----------|
| Baseline| 15.28 | 27.64    | 13.51    | 53.64    | 68.21    |
| Phase B | 7.24  | 7.92     | 4.64     | 48.98    | 54.72    |
| Phase C | 2.38  | 4.86     | 1.66     | 42.4     | 46.68    |

Table 10 demonstrates performances of ActSCA when attacking other 4 CRYTAL-Kyber target devices using 10 attack traces. Though mean-ranks unsuprisingly vary on different devices due to hardware/software variations, ActSCA consistently helps to boost performances of Baseline considerably. This demonstrates the generability and effectiveness of ActSCA in cross-device settings.

# G    PERFORMANCES OF ACTSCA ON AES CRYPTOGRAPHY

Table 11: Medians of mean ranks over 100 runs for ActSCA on different AES devices using 10 attack traces. Best results are highlighted in bold.

|         | Same | Device 1 | Device 2 | Device 3 | Device 4 | Device 5 |
|---------|------|----------|----------|----------|----------|----------|
| Baseline| 3    | 16.4     | 7        | 216.6    | **1**    | 33.8     |
| Phase B | 2.9  | 35.6     | 4        | 119      | 3        | 18.1     |
| Phase C | **2.8** | **20.2** | **3**    | **87.2** | **1**    | **14.5** |

**Unprotected AES.** Table 11 reports the mean-ranks of an unprotected 128-bit AES implementation (Boyang & Mabon) across five different devices using only 10 attack traces (data provided in (Boyang & Mabon)). ActSCA (Phase B and Phase C) consistently outperforms the Baseline, demonstrating significant improvements in cross-device generalization. Notably, Phase C further refines performance over Phase B, especially on challenging targets like Device 3 (118 to 86.2) and Device 5 (17.1 to 13.5), indicating the effectiveness of the final adaptation step. These results confirm that ActSCA can provide more stable and accurate attacks across heterogeneous devices compared to traditional approaches.

Table 12: Medians of mean ranks over 100 runs on the ASCAD protected device for ActSCA and other SOTA methods. Best results are highlighted in bold.

| Method   | 500 Trace | 800 Trace |
|----------|-----------|-----------|
| ActSCA   | **86.5**  | **116.5** |
| Baseline | 161.5     | 168       |
| MMD      | 126.5     | 199       |
| ADA      | 209       | 185.5     |
| ZMUV     | 152       | 173       |
| MDMSD    | 136.5     | 141.5     |

**Protected AES from the ASCAD dataset.** We additionally evaluate performances of ActSCA on protected devices obtained from the ASCAD dataset (Emmanuel et al., 2018). The ASCAD

dataset includes electromagnetic (EM) or power traces captured from 8-bit AVR microcontrollers (ATmega8515) running masked AES-128 implementations (with first-order boolean masking at the SubBytes level for protection). Tables 12 below show medians of mean ranks over 100 runs of ActSCA and SOTA methods on protected devices from ASCAD using 500 and 800 attack traces. ActSCA outperforms other SOTA models significantly (up to 2.4x).

Table 13: Median mean ranks of ActSCA over 100 runs with different distance metrics on the protected device ASCAD compared to the baseline.

| ASCAD | Baseline | L2+KLL | L2+$\alpha$-KLL | L1+KLL | L1+$\alpha$-KLL | Cos+KLL | Cos+$\alpha$-KLL |
|---|---|---|---|---|---|---|---|
| #500 traces | 161.5 | 86.5 | 96.5 | 87.5 | 80.5 | 90.5 | 85.5 |
| #800 traces | 168 | 116.5 | 109.5 | 110.5 | 113.5 | 119.5 | 115.5 |

**Robustness of ActSCA on distance functions.** In Table 13, we show the medians of mean ranks over 100 runs and 500 attack traces of ActSCA on the protected device ASCAD using some distance functions including $L_1$, Cosine (Cos), and $\alpha$-KLL (Kimura & Hino, 2021) besides $L_2$+KLL in our paper (cf. Section 2.3 for details). Though the results vary slightly with different metric combinations, our method AcSCA consistently outperforms the baseline in all cases. These show that ActSCA is quite robust to the choices of metrics even when dealing with protected devices.

Table 14: Median mean ranks of ActSCA with different baselines on ASCAD over 100 runs and 500 attack traces. The median mean ranks for SOTA methods are: 126.5 (MMD), 209 (ADA), 152 (ZMUV) and 136.5 (MDMSD).

| | Without ActSCA | With ActSCA |
|---|---|---|
| EFCNN | 123 | **32** |
| InceptionNet | 244 | **208** |
| CNN_best | 126 | **74** |
| MLP | 168 | **116.5** |

**Performance of ActSCA with different baselines.** Table 14 shows the medians of mean ranks of ActSCA when using with different baselines (base models), including MLP, CNN_best model from ASCAD paper (Emmanuel et al., 2018), InceptionNet (Huang et al., 2025) and EFCNN (Liu et al., 2025), over 100 runs with 500 attack traces. As we can see, ActSCA can significantly boost performance of these base models up to 3.84x. These demonstrate its generality when using with arbitrary base models. Moreover, compared to SOTA methods including MMD, ADA, ZMUV and MDMSD, ActSCA with EFCNN acquires up to 6.5x mean-rank improvement. Hence, the stronger the baselines, the better the performance of ActSCA.

Table 15: Medians of mean ranks over 100 runs on the AESptv2 protected devices D1-D4 for ActSCA and other SOTA models. Best results are highlighted in bold.

| #Trace | Method | D1 | D2 | D3 | D4 |
|---|---|---|---|---|---|
| 500 | ActSCA | **55.5** | **115.5** | **65.5** | **78** |
| | Baseline | 80 | 151.5 | 99.5 | 131 |
| | ADA | 198 | 178.5 | 237 | 199.5 |
| | MMD | 120 | 165 | 143.5 | 202 |
| | ZMUV | 96.5 | 187.5 | 104 | 152.5 |
| | MDMSD | 137 | 126.5 | 112.5 | 115 |
| 800 | ActSCA | **46** | **85** | **104** | **58** |
| | Baseline | 72 | 162.5 | 151.5 | 69 |
| | ADA | 203 | 105.5 | 165 | 210.5 |
| | MMD | 156 | 202 | 210 | 156 |
| | ZMUV | 84 | 160 | 161.5 | 72.5 |
| | MDMSD | 181.5 | 178 | 167.5 | 104 |

**Protected AES from the AES_PTv2 datasets.** Since the ASCAD dataset only provides 1 device, to further demonstrate the effectiveness of ActSCA, we study the AES_PTv2 dataset, a recent and

comprehensive benchmark developed to evaluate the portability and generalization of side-channel attacks across multiple devices and countermeasure implementations (Rioja et al., 2021). It includes side-channel traces collected from 4 physical devices D1-D4 using four STM32F411VE boards, covering three software implementations of AES-128: unprotected, Masking Scheme 1 (MS1), and Masking Scheme 2 (MS2). MS1 applies a mask per S-Box lookup and removes it immediately after, while MS2 applies masking throughout SubBytes. Since the Masking Scheme 2 operates at SubBytes level, similar to ASCAD, we decide to use Masking Scheme 1 to show that our algorithm can work with a variety of protection schemes. We use the same experiment settings like ASCAD. The results are demonstrated in Table 15 with 500 and 800 attack traces over 100 runs. ActSCA dramatically outperforms all existing SOTA models with up to 3.6x mean rank improvements.

## H  RELATED WORKS (EXTENDED)

**ActSCA vs. Data selection methods.** There are various data selection methods proposed in the literature. For example, JTT (Liu et al., 2021) improves the models via two training stages to identify hard samples to upweighting them before retraining the model. DynUn (Tan et al., 2025) select samples based on the model training dynamics while InfoMax utilize graph information to select appropriate samples. Influence functions based approaches such as (Banerjee et al., 2024) estimate the importance of training data points and use them to improve model performance by down- or up-weighting samples for retraining. They use all the training data, which is different to ActSCA which uses only a part of the training data.

**ActSCA vs. Active Learning (AL).** In traditional active learning approach (Settles, 2009), a ML model $M$ has access to a pool of unlabeled data $U$ so that it can iteratively take some data to ask for labels and use them to update $M$. Dual-Leak (Yu et al., 2023) discussed above is an example of traditional methods. Though Phase B of ActSCA resembles AL, it differs fundamentally in following points. First, at each iteration, the data (traces) we select have their own labels ($D_P$ is a labeled pool) and their labels will also be used to assess their selection qualifications. Second, a trace can be re-selected at different iterations.

**ActSCA vs. Transfer Learning.** Similarly, Phase C of ActSCA may resemble Transfer Learning since we have the model $M_2$ to be fine-turned on a dataset $D_T$ to create a new model $M_3$. However, compared to other transfer learning approaches like (Yu et al., 2021), $D_T$ is very small (can be a single or a few samples) and is completely unlabeled. Moreover, $D_T$ is totally not used for training $M_3$. Instead, it is used for obtaining a dataset $D_3$ from the source domain to update the source model $M_2$.

## I  MORE DETAILS ON ACTSCA

**More details on ActSCA framework.** Our ActSCA framework ilustrated in Figure 2 consists of 3 Phases: (A) Data representation selection, (B) Active model updating and (C) Cross device adjustment. Readers may also find the high-levl pseudo code in Algorithm 2. During the profiling stage, in Phase A, all data in are first grouped into homogeneous clusters using k-medoids. All cluster centroids are selected as an initial data representation set (c.f. Section 2.1 Data clustering and Sample Selection). Then, Min-max queries are continuously executed to construct the full data representation set containing points that diversely represent the entire data space. DL model is trained with as an initial model to construct updated model in Phase B (c.f. Section 2.1 Balanced Max-min Sampling). In Phase B, iteratively and actively selects subsets of data that it thinks most important from (c.f. Section 2.2 Active data selection) and updates itself to gradually improve its performance (c.f. Section 2.2 Active model updating). At the end of Phase B, is expected to have better performance than training with full and will be used as a base model for attacking target devices. However, in the attacking stage, if using directly, we cannot exploit characteristics of the target device effectively to improve performance. Hence, in Phase C, ActSCA finds a subset of data that match well with from both data and model perspectives (c.f. Section 2.3 Similarity Matching) and uses it to create a model tailored specifically for for further performance boost (c.f. Section 2.3 Target device specific model creation).

**Input** : Profiling pool $\mathcal{D}_P = \{(x_i, y_i)\}$; target-device traces $\mathcal{D}_T = \{x_j^{(T)}\}$;
number of Phase B iterations $N_B$;
BMMS selector $\texttt{BMMS}(\cdot)$ (Eq. (2)); BUS selector $\texttt{BUS}(\cdot)$ (Eq. (5));
similarity metrics: prediction KL (Eq. (6)), feature distance (Eq. (7));
support-set combiner (Eq. (8)).
**Output:** Final model $M_3$

**Phase A: Data Representation Selection**               `// Initial data selection`
$\mathcal{C} \leftarrow \texttt{KMedoids}(\mathcal{D}_P)$                              `// cluster` $\mathcal{D}_P$ `to get medoids`
$\mathcal{D}_1 \leftarrow \texttt{MedoidsAsSet}(\mathcal{C})$                        `// initial representative set`
$\mathcal{D}_1 \leftarrow \texttt{BMMS}(\mathcal{D}_P, \mathcal{D}_1)$                                        `// Eq. (2)`
$M_1 \leftarrow \texttt{Train}(\mathcal{D}_1)$                                  `// train initial model`

**Phase B: Active Model Updating**               `// Active model updating`
$M_2 \leftarrow \texttt{InitFrom}(M_1)$
**for** $t \leftarrow 1$ **to** $N_B$ **do**
  $\mathcal{S}_t \leftarrow \texttt{BUS}(M_2, \mathcal{D}_P)$      `// select informative samples via Eq. (5)`
  $\mathcal{S}_t \leftarrow \mathcal{S}_t \cup \texttt{AugmentWithClusterNeighbors}(\mathcal{S}_t, \mathcal{D}_P)$
  $\mathcal{D}_1 \leftarrow \mathcal{D}_1 \cup \mathcal{S}_t$
  $M_2 \leftarrow \texttt{Finetune}(M_2, \mathcal{D}_1)$
**end**

**Phase C: Cross-Device Adjustment**               `// Target device adjustment`
$\mathcal{D}_3 \leftarrow \varnothing$
**foreach** $x^{(T)} \in \mathcal{D}_T$ **do**
  $\mathcal{N}_{\text{pred}} \leftarrow \texttt{RetrieveByPredKL}(M_2, x^{(T)}, \mathcal{D}_P)$                         `// Eq. (6)`
  $\mathcal{N}_{\text{feat}} \leftarrow \texttt{RetrieveByFeatDist}(M_2, x^{(T)}, \mathcal{D}_P)$                       `// Eq. (7)`
  $\mathcal{S}(x^{(T)}) \leftarrow \texttt{Combine}(\mathcal{N}_{\text{pred}}, \mathcal{N}_{\text{feat}})$                              `// Eq. (8)`
  $\mathcal{D}_3 \leftarrow \mathcal{D}_3 \cup \mathcal{S}(x^{(T)})$
**end**
$M_3 \leftarrow \texttt{FreezeFeatureExtractor}(M_2)$                         `// freeze backbone`
$M_3 \leftarrow \texttt{FinetuneHead}(M_3, \mathcal{D}_3)$                      `// update classifier only`

**return** $M_3$

**Algorithm 2:** High Level Pseudo code for ActSCA

**Input** : Data Pool $D_P$, number of clusters $k$, number of core-set instances $\delta$
**Output:** Set $D_1$ that contains core-set instances
$S = \text{K-Medoids}(D_P, k)$;
$D_1 = S$;
**while** $|D_1| < \delta$ **do**
  $d_X = \emptyset$;
  **foreach** *instance* $\boldsymbol{x} \in D_P \backslash D_1$ **do**
    $d_X.\text{insert}(\min d(\mathbf{x}, D_1))$
  **end**
  $\mathbf{x} = d_X.\text{argmax}()$;
  $D_1.\text{insert}(\mathbf{x})$;
**end**
**return** $D_1$ ;

**Algorithm 3:** Balanced Max-Min Sampling Algorithm

**Pseudo-codes for ActSCA.** Algorithms 3, 4 and 5 shows pseudo-codes for the three main Phases A, B, and C of ActSCA described in Section 2, respectively. High-level pseudocodes for ActSCA can also be found in Algorithm 2.

**Input**  : Data pool $D_P$, number of iteration $N_{it}$, number of sampling instances $\epsilon$, model $M_1$
**Output:** Trained model $M_2$
$\eta = 0$;
**while** $\eta < N_{it}$ **do**
    $D_2^\eta = $ Uncertainty Balance Sampling$(D_P, \epsilon)$;
    $D_{2u}^\eta = D_2^\eta + C(D_2^\eta)$;
    Update$(M_1, D_{2u}^\eta)$;
    $\eta = \eta + 1$;
**end**
$M_2 = M_1$;
**return** $M_2$;

**Algorithm 4:** Active Model Updating Algorithm

**Input**  : Data pool $D_P$, target device data $D_T$, neighborhood size $\kappa$, regulation parameter $\beta$,
        model $M_2$
**Output:** Trained model $M_3$
$D_3 = \emptyset$;
/* Procedures for selecting samples from $D_P$ by the metric $d_P$ */
$d_{KL} = \emptyset$;
**foreach** $\boldsymbol{x}_p \in D_P$ **do**
    $d_{KL}$.insert$(d_P(\mathbf{x}_p, D_T))$;
**end**
$D_3 = D_3$.union$(d_{KL}$.argmax$(\kappa \cdot \beta))$;
/* Procedures for selecting samples from $D_P$ by the metric $d_F$ */
$d_{KNN} = \emptyset$;
**foreach** $\boldsymbol{x}_p \in D_P$ **do**
    $d_{KNN}$.insert$(d_F(\mathbf{x}_p, D_T))$;
**end**
$D_3 = D_3$.union$(d_{KNN}$.argmax$(\kappa \cdot (1 - \beta)))$;
**for** $N_{epoch}$ **do**
    Fine-tune$(M_2, D_3)$;
**end**
$M_3 = M_2$;
**return** $M_3$;

**Algorithm 5:** Cross Device Adjustment Algorithm

**Proof for Theorem 1.** Before going into our proof, we present Claim 1 from (Berlind & Urner, 2015). For any Bernoulli distribution $p, p' \in [0, 1]$ and $y' \in \{0, 1\}$,

$$\mathbb{P}_{y \sim p}(y \neq y') \leq \mathbb{P}_{y \sim p'}(y \neq y') + |p - p'| \tag{9}$$

Let $\mathbf{x}_i \in \mathcal{D}_\mathcal{P}$ and $\mathbf{x}_j \in S$. Since $S$ is constructed using $k$-medoids to get $k$ clusters, we have $S = \{\mathbf{s}_1, \mathbf{s}_2, \cdots, \mathbf{s}_k\}$. Let $C_k = \{\mathbf{s}_k, \mathbf{x}_1, \mathbf{x}_2, .., \mathbf{x}_{N_k}\}$ be the cluster that contains $N_k + 1$ number of points including medoid $\mathbf{s}_k$. With each cluster, we can represent the set of distance between $\mathbf{s}_k$ and member points $\mathbf{x}_1, \mathbf{x}_2, .., \mathbf{x}_{N_k}$ as $dist(\mathbf{s}_k, \mathbf{x}_i)$ for $i = 1, 2, ..n$. Denotes $R_k = max_{i=1}^{N_k} dist(\mathbf{s}_k, \mathbf{x}_i)$ as the largest distance between its medoid $\mathbf{s}_k$. The distance between medoid set $S$ and its cluster members is $R = max_{k=1}^{k} R_k$ and we can bound the Euclidean distance between $\mathbf{x}_i \in D_P$ and $\mathbf{x}_j \in S$ where $\mathbf{x}_i$ and $\mathbf{x}_j$ are in the same cluster as $||\mathbf{x}_i - \mathbf{x}_j|| \leq R$.

Let $\eta_k(\mathbf{x}_i) = \mathbb{P}(y_i = k \mid \mathbf{x}_i)$ denote the true class probability for class $k$ given $\mathbf{x}_i$, and assume the ground-truth label $y_i \sim \eta(\mathbf{x}_i)$. Then the expected loss on $\mathbf{x}_i$ is:

$$\mathbb{E}_{y_i \sim \eta(\mathbf{x}_i)}[l(\mathbf{x}_i, y_i; M_1)] = \sum_{k=1}^{|\mathcal{K}|} \eta_k(\mathbf{x}_i) l(\mathbf{x}_i, k; M_1)$$

Using the triangle inequality from Equation 9, with $l(\mathbf{x}_i, k; M_1) \geq 0$ we obtain:

$$\sum_{k=1}^{|\mathcal{K}|} \eta_k(\mathbf{x}_i)l(\mathbf{x}_i, k; M_1) \leq \sum_{k=1}^{|\mathcal{K}|} \eta_k(\mathbf{x}_j)l(\mathbf{x}_i, k; M_1) + \sum_{k=1}^{|\mathcal{K}|} |\eta_k(\mathbf{x}_i) - \eta_k(\mathbf{x}_j)|l(\mathbf{x}_i, k; M_1)$$

We use the $K^\mu$-Lipschitz property of the regression function $\eta_k(.)$ and get the following inequalities:

$$|\eta_k(\mathbf{x}_i) - \eta_k(\mathbf{x}_j)| \leq K^\mu \|\mathbf{x}_i - \mathbf{x}_j\| \leq K^\mu R$$

Since the loss is bounded $l(\mathbf{x}_i, k; M_1) \leq L$, we obtain the following inequality for the second term:

$$\sum_{k=1}^{|\mathcal{K}|} |\eta_k(\mathbf{x}_i) - \eta_k(\mathbf{x}_j)|l(\mathbf{x}_i, k; M_1) \leq RK^\mu L \sum_{k=1}^{|\mathcal{K}|} 1 = RK^\mu L|\mathcal{K}|$$

Now for the first term, since the model is trained to zero training loss on the selected subset $s$, i.e., $l(\mathbf{x}_j, k; M_1) = 0$, we assume (due to $K_l$-Lipschitz continuity of the loss):

$$|l(\mathbf{x}_i, k; M_1) - l(\mathbf{x}_j, k; M_1)| \leq K^l \|\mathbf{x}_i - \mathbf{x}_j\| \leq K^l R \Rightarrow l(\mathbf{x}_i, k; M_1) \leq K^l R$$

So we get:

$$\sum_{k=1}^{|\mathcal{K}|} \eta_k(\mathbf{x}_j)l(\mathbf{x}_i, k; M_1) \leq RK^l$$

Putting everything together:

$$\mathbb{E}_{y_i \sim \eta(\mathbf{x}_i)}[l(\mathbf{x}_i, y_i; M_1)] \leq RK^l + RK^\mu L|\mathcal{K}| = R(K^l + K^\mu L|\mathcal{K}|)$$

Finally, using Hoeffding's inequality over $N_P$ samples with bounded loss in $[0, L]$, with probability at least $1 - \xi$, we have:

$$\left| \frac{1}{N_P} \sum_{(\mathbf{x}_i, y_i) \in D_P} l(\mathbf{x}_i, y_i; M_1) - \frac{1}{N_s} \sum_{(\mathbf{x}_j, y_j) \in S} l(\mathbf{x}_j, y_j; M_1) \right| \leq R(K^l + K^\mu L|\mathcal{K}|) + \sqrt{\frac{L^2 \log(1/\xi)}{2N_P}}$$

This completes the proof.

## J  BROADER IMPACT

**Societal Impact.** ActSCA is designed to advance the evaluation of cryptographic implementations by improving the efficiency and effectiveness of side-channel attacks, particularly in realistic cross-device settings. The societal benefit of this work lies in its potential to strengthen the security of cryptographic systems by providing security evaluators and hardware designers with a more powerful and adaptive tool for identifying implementation vulnerabilities. By exposing weaknesses that traditional methods may overlook, ActSCA contributes to the design of more resilient cryptographic protections. However, as with many advancements in offensive security, there exists a risk that malicious attacker could exploit these techniques to bypass cryptographic systems. To mitigate this risk, we disclose our findings responsibly and emphasize that the primary goal of ActSCA is defensive: to support the development of robust, leakage-resilient cryptographic systems through better empirical evaluation. We encourage the use of ActSCA within a controlled, ethical, and research-oriented context to maximize its positive societal impact while minimizing misuse.

**Ethical Impact.** The ethical implications of this research were considered at the start and throughout this project. The research designs and evaluates techniques to infer cryptographic keys using side channel information. This research is motivated by the need to properly test the robustness of encryption methods. As such, the main positive outcome is to inform the design of safe and robust cryptographic techniques. The research however has a potential negative impact where its outcome

may be used practically by malicious actors before corrective actions are undertaken. The ethical considerations for this project are detailed below.

**Respect for Persons**: This study does not involve the participation of human research subjects.

**Beneficence**: We base our considerations on the principle of maximizing probable benefits while aiming to minimize probable harms. We find that the probable benefits outweigh the probable harms because of the following: It is important to understand the robustness of cryptographic methods. Investigating methods to infer cryptographic keys is an important and widely practiced research activity with the purpose of discovering vulnerabilities early and to maximize the security of practical designs. A negative side effect of this research is that techniques are developed and shared that can potentially be used by malicious actors. However, there is a broader consensus that public knowledge of vulnerabilities, shared at the right time to allow corrective actions, is in the public interest. The presented research has been conducted in this context, with the intent to disclose exploitable vulnerabilities using proper processes. This way, the impact on users, developers of cryptographic systems and related companies is balanced.

**Justice**: From an equity standpoint, we have considered the positive and negative impact on users and developers in light of the potential identification of vulnerabilities. Our consideration is that the research, as conceived and executed, advances understanding of side channel attacks, however is of such a nature that it does not dramatically change the balance of the burden on users and developers.

**Respect for Law and Public Interest Transparency and Accountability, and Compliance**: The processes behind this research are fully documented and artifacts are shared to the extent permitted by law.

## K   LIMITATIONS

Despite its effectiveness, the proposed method presents several limitations. First, the number of target traces $D_T$ is fixed in our current setting, making it unclear how many traces are minimally required to guarantee a successful attack in practice. Second, while ActSCA involves a small number of hyperparameters that remain stable across experiments (c.f Section D), a parameter-free variant may offer a more robust and user-friendly alternative—this is left as future work. Third, regardless o the selection strategies we use in Phase A and B of ActSCA, bad samples may still slip through. Hence, having more and more effective selection strategies are a tempting target in the future.

## L   LLM USAGE

This manuscript was very slightly edited using LLMs for language polishing and writing improvements. The authors retain full responsibility for the research content, including the concepts, analyses, and conclusions.

