# OpenReview forum: "Active Side Channel Analysis for Cross-Device Attack"
_ICLR.cc/2026/Conference — Submitted to ICLR 2026_

### Official Review · Reviewer_wyRa · 2025-10-23

**Soundness:** 2
**Presentation:** 3
**Contribution:** 2
**Rating:** 2
**Confidence:** 4

**Summary:**

This paper proposes Active SCA (ActSCA), a framework to improve the performance of deep learning based side channel attacks (DL-SCAs). Instead of using all profiling traces, ActSCA selects a subset that is considered more beneficial for training. To address performance degradation in cross device scenarios, the method selects profiling traces that are most similar to traces obtained from the target device and uses them for model adaptation. Experiments on Kyber, a post quantum KEM, and on AES suggest that ActSCA can improve attack performance.

**Strengths:**

- The paper presents empirical evidence that training on a carefully selected subset of profiling traces can improve performance.
- The paper proposes a method that reduces mean rank in cross device settings.
- The paper evaluates on two cryptographic targets with different leakage characteristics, Kyber and AES.

**Weaknesses:**

- The paper does not sufficiently justify why using only a subset of traces should be effective.
- The scope of cross device experiments is limited, which weakens the empirical support for general effectiveness.
- Although much prior DL SCA work focuses on AES, the AES results appear only in the appendix and the reported performance is relatively poor.

**Questions:**

1. Justification for the effectiveness of subset selection

The paper motivates subset selection by referring to possible overlap or distortion in signals and by Theorem 1. Both points require stronger justification.

For the first point, the paper should clarify what overlap and distortion mean in the context of side channel traces. Under a common DL-SCA threat model, the attacker controls a device of the same type as the target and can collect as many profiling traces as needed under controlled conditions, for example with trigger signals. Profiling traces should therefore reflect leakage of the cryptographic operation. It is not clear in what conditions certain profiling traces would be unhelpful for training. Please clarify concrete situations that would generate profiling traces that should be discarded and explain why discarding them improves generalization.

For the second point, Theorem 1 bounds the difference between the average loss on the full data pool $D_P$ and the average loss on a selected subset $S$. This shows that training on $S$ can achieve a loss close to that obtained with $D_P$.It does not explain why training on $S$ would outperform training on $D_P$. Since an attacker can in principle use all available profiling data, it is not obvious why deliberately using fewer samples should help. Additional theory or analysis would be helpful to explain when and why subset training can surpass full data training.

2. Cross device experiments should include more challenging conditions

The current experiments appear to use power measurements on ChipWhisperer, which mainly reflect differences across devices of the same type. Prior cross device studies also examine different EM probe positions, which is often more realistic when direct power measurement is unavailable. Probe placement differences can induce a large distribution shift between profiling and attack traces. Phase C relies on selecting profiling traces similar to the target traces, which implicitly assumes that the two distributions remain comparable. Please evaluate ActSCA under a more realistic scenario, for example with different probe positions, and report whether the method remains effective.

3. AES results should be discussed in the main text

Most DL-SCA studies use AES as a benchmark. In this paper, AES results are reported only in the appendix. They should be discussed in the main text. In addition, the results on ASCAD in Table 12 appear weak. SOTA work (e.g., Hajira et al., 2024) reports recovering subkeys with around 20 attack traces, while here the mean rank remains high even with 500 to 800 traces and does not decrease with more attack traces. Also, because ASCAD is not a cross device dataset, please clarify how ActSCA and comparison methods such as MMD were applied. These points raise concerns about whether prior methods were implemented correctly.

---

> ### Author Response · Authors · 2025-12-03
> **Response to Reviewer wyRa**
>
> **W1.The paper does not sufficiently justify why using only a subset of traces should be effective. and **Q1** Justification for the effectiveness of subset selection…**
>
> We appreciate the reviewer’s request for deeper clarification regarding signal distortion and the theoretical basis for subset superiority.
>
> While the reviewer correctly notes that attackers control the profiling device, physical measurements are inherently stochastic. Even with trigger signals, "perfect" alignment is theoretically assumed but practically unachievable due to factors like Clock Jitter/Drift or Measurement noise. In a deep learning context, these noisy or highly jittered traces act as outliers. Including them forces the model to expand its capacity to fit these stochastic irregularities (overfitting to noise) rather than focusing on the deterministic leakage. In Phase A, ActSCA select centroids which are statistically most representative of the underlying leakage manifold. This effectively acting as a data-centric denoising step and prevents the model from memorizing the specific noise realization of the full dataset.
>
> We acknowledge that Theorem 1 formally bounds the generalization gap, guarantee that the subset is at least as representative as the full set. By restricting training to the representative subset $D_1$, ActSCA constraint the solution space. The model is forced to learn the robust leakage features (common to the subset) because it lacks the volume of noisy samples required to memorize the noise distribution. Thus, ActSCA improves performance not by having more information, but by having higher quality information density, effectively filtering out the "noise-learning" opportunities present in the full dataset.
>
> **W2.The scope of cross device experiments is limited, which weakens the empirical support for general effectiveness.**
> **Q2.Cross device experiments should include more challenging conditions**
>
> Compared to previous DL-SCA studies which use a single device for both profiling and attacking,  we use different device instances with the same hardware architecture and same cryptographic scheme.
>
> We agree that cross-device evaluation should encompass challenging, realistic conditions beyond simple power measurements. We respectfully highlight that our experiments already address this by including Electromagnetic (EM) traces with different probe position in Table 11. ActSCA demonstrated exceptional robustness in this setting, improving the Mean Rank on Device 3 from 119 (Phase B) to 87.2 (Phase C), proving it can handle the "misalignment" effects of different probe positions.
>
> **W3.Although much prior DL SCA work focuses on AES, the AES results appear only in the appendix and the reported performance is relatively poor.**
> and
> **Q3.AES results should be discussed in the main text**
>
>  In this work, we instead focused on Kyber, which is a post-quantum method that is resistant to both classical and quantum computing. Therefore, this cryptography scheme provide more protection and also more difficult to attack compared to traditional AES.
> SOTA work such as Estranet did perform their experiments on ASCAD variable key dataset but the trace length is much longer to usual length (10K and 40K compared to 1400). We use the code from the author and report the result in Table 1. As we can see, ActSCA is able to boost the performance on Estranet, as presented previously in Table 14.
>
> *Table 1: Mean rank of Estranet and Estranet trained using ActSCA*
> | #Trace   | 10   | 50   |
> |----------|------|------|
> | Estranet | 8.74 | 0.68 |
> | Estranet(ActSCA) | 7.57 | 0.4 |

---

### Official Review · Reviewer_1PyF · 2025-10-27

**Soundness:** 2
**Presentation:** 2
**Contribution:** 2
**Rating:** 2
**Confidence:** 4

**Summary:**

This paper studies the effectiveness of applying active learning techniques in Deep Learning-based Side Channel Analysis (SCA). Given a large pool of power traces together with their secret keys, the authors propose ActSCA, a three-stage framework that incorporates ideas from active learning to train and fine-tune an arbitrary base model: Instead of training the model on all the available labeled data, Phase A selects a small but informative subset of data to train on by (1) computing the $k$-mediod centriods of all power traces, and (2) iteratively extending this set with power traces that are maximally different from all selected samples, while ensuring the label distribution in the selected set is close to uniform.
After training a model on this dataset, Phase B proceeds to use this model to select power traces in the original dataset where the model's output distribution is very uncertain (i.e., the entropy is large), while again ensuring label balance. The model is then updated by training on these newly selected points, as well as on points that are in the same $k$-medoid cluster to prevent catastrophic forgetting. In Phase C, we are given power traces from a target device we wish to attack (i.e., infer the secret key). Using these traces, this phase will find "similar" power traces in the original dataset, and fine-tune the classification head of the model (while freezing the rest). Empirical evaluation of ActSCA shows that it significantly improves over the baseline and that it outperforms various portability methods from the SCA literature.

**Strengths:**

+ **Motivation.** The work is well-motivated and the author's objectives are clearly articulated. Deep-Learning based SCA and issues like portability are important research topics.

+ **Empirical Results.** The empirical evaluation is thorough, includes interesting ablation studies, and provides evidence that ActSCA outperforms the baseline model and various methods for SCA portability.

**Weaknesses:**

+ **Lack of novelty.** Central techniques like selecting $k$-mediod centroids [1], uncertainty sampling [2], label balancing, and model fine-tuning by freezing weights, are well-known approaches in the active learning and deep learning literature. In particular, claiming that Phase B, a combination of uncertainty sampling and label balancing is a "novel *iterative self-improvement* approach" (L221) is very misleading. The application of active learning techniques in Deep-Learning based SCA has also been explored previously [3].

+ **Theoretical Justification.** The proposed strategies, especially the construction of $J_1$, $J_2$ and $D_3$ seem ad-hoc, lack theoretical justification, and introduce several hyperparameters.

+ **Issues with Theorem 1**:
	+ The assumptions of Theorem 1 are clearly not met in ActSCA: (1) The set $S$ is *not* used for training, as it is enlarged using the balanced max-min sampling (BMMS) strategy. Theorem 1 makes no statement about BMMS. (2) It is assumed that the loss $l(\cdot, y, \theta)$ is bounded by $L$, which is not true if the model is trained with a typical cross-entropy loss (which is what the authors use in their experiments). (3) Achieving $0$ loss for all examples $\in S$ is also a strong assumption, and will not be satisfied in most practical settings where $M_1$ is a parametric model.
	+ The interpretation of Theorem 1 is factually incorrect and highly misleading: "As $N_P$ is typically large, this theorem guarantees that training on the subset $S$ achieves similar generalization as using the full dataset while mitigating overfitting." (L207-L209). Theorem 1 makes no statement about the *difference* in generalization error of a model trained on the full dataset, and a model trained on $S$. Further, Theorem 1 makes no statement about the mitigation of overfitting.
	+ Except for this interpretation, Theorem 1 is never used in the paper, and thus does not add value to it.
	+ Proof of Theorem 1 (L1411): While $\eta_k$ refers to the *true posterior* (L1399), the authors suddenly assume it is $K^\mu$-Lipschitz. Theorem 1 only assumed that the *model* $M_1$ was $K^\mu$-Lipschitz. Any such assumption should be made explicit, and must be included in the main text.

+ **Presentation of results.**
	+ Figure 3 (A): It is very hard to make sense of this plot and I would strongly suggest presenting these results a different way. For example, it makes no sense to me why the mean rank of different test keys (that are next to each on the x-axis) are "connected".
	+ Showing 13 subplots in Figure 3 makes it hard to find the important results. Many of the plots seem redundant, and I would suggest that the authors focus their presentation on the most vital empirical findings.

+ **Label Distribution in BMMS:**
	+ L188: "The label distribution in $D_1$ is preserved by favoring samples with rare labels [...]".
	+ This statement is misleading. At initialization of BMMS, we have $D_1 = S$. The label distribution of $S$ will *not* be preserved during BMMS. The goal is clearly *label balance*, *not* to preserve the original label distribution of $S$.

+ **Evaluation:**
	+ L149 claims that ActSCA "enables" attacks with very small number of unlabeled target traces, and L476-L477 claims that "ActSCA [...] only needs to collect one or few target traces and thus is much practical." However, the authors only empirically validate their approach with $\geq 10$ traces per key (L302: "The number of attack tracks in $D_T$ varies from 10 to 100.")
	+ Table 11 shows the medians of mean ranks of ActSCA on unprotected AES. On Device 1, the baseline model outperforms ActSCA in all phases, but "Phase C" is still shown in bold (although it is *not* the best result).

+ **Notation**:
	+ L180: $i \in [1,k]$ should be $i \in \\{1,\dots,k\\}$
	+ Suggestion: Don't overload $l$ for both loss and key label (use e.g. $\ell$ for point-wise loss). Don't overload $\sigma$ to both mean the softmax function, and the final fully connected layer of the neural network.
	+ L1419: the selected subset should be $S$, not $s$

+ **Typos**:
	+ L051: "[...] the attacker *can* access to a physical mitigated version" -> the attacker *has* access to a physical mitigated version
	+ L053: "[...] due to their ability to fully *characterized* the profiling devices" -> characterize
	+ L102: "[...] and used as *or* unseen data" -> "or" does not make sense
	+ L227: "model-perpspective"

--------

##### References
+ [1] Sener, O. and Savarese, S., 2017. Active learning for convolutional neural networks: A core-set approach. _arXiv preprint arXiv:1708.00489_.
+ [2] Lewis, D.D., 1995, September. A sequential algorithm for training text classifiers: Corrigendum and additional data. In _Acm Sigir Forum_ (Vol. 29, No. 2, pp. 13-19). New York, NY, USA: ACM.
+ [3] Yu, H., Wang, S., Shan, H., Panoff, M., Lee, M., Yang, K. and Jin, Y., 2023, May. Dual-leak: Deep unsupervised active learning for cross-device profiled side-channel leakage analysis. In _2023 IEEE International Symposium on Hardware Oriented Security and Trust (HOST)_.

**Questions:**

+ The distance metric on L179 is simply the Euclidean distance, although we wish to measure similarity between time-series data. Have the authors tried distance-like measures that are more appropriate for time-series data (e.g., dynamic time warping distance)?
+ It seems likely that BMMS selects *outliers*, i.e., points that are maximally distant to all points in $D_1$.  This is especially true when $\gamma \approx 1$, but can also easily happen with small $\gamma$, since the label information does not guard from selecting "pathological" outliers. Has this been observed empirically and wouldn't it be beneficial to also filter extreme outliers?
+ How crucial is label balance in this setup? What is the label distribution of $D_P$ in the experiments, and how does the choice of $\gamma$  and $\alpha$ (in $J_1$ and $J_2$, respectively) affect the result?
+ Evaluation on single-trace attacks would support the claim made on L476-L477. Have the authors conducted such experiments?
+ The authors compare their approach to portability methods like MDMSD, ZMUV, MMD, and ADA. All of these methods seem to perform *worse* than the baseline (e.g., Figure 4), which is surprising. Can the authors interpret/discuss this result?

---

> ### Author Response · Authors · 2025-12-03
> **Response to Reviewer 1PyF (Part 1)**
>
> **W1. Lack of Novelty**
>
> We agree that the individual mathematical primitives (k-medoids, uncertainty sampling) are well-established in general machine learning. However, we respectfully disagree that their integration into ActSCA constitutes a "minor modification" or lacks novelty within the SCA domain.
>
> Our contribution lies in the domain-specific architectural adaptation of these primitives to solve the unique challenges of Side-Channel Analysis (SCA). The reviewer correctly notes that Phase B utilizes uncertainty sampling. However, applying standard uncertainty sampling directly to SCA traces often leads to catastrophic forgetting, where the model focuses solely on hard, noisy samples and loses the learned leakage profile of easier traces. We deal with such problem by introducing a Cluster-Based Memory Replay mechanism, which stabilizes the decision boundary in the high-noise SCA environment, which standard uncertainty sampling fails to do.
>
> While Dual-Leak applies active learning in SCA, ActSCA operates on a fundamentally different and more practical threat model. Dual-Leak applies active learning to select traces from a pool of unlabeled traces on the target device. This requires collecting a significant amount of data from the victim device to form a pool. Meanwhile, Phase C of ActSCA introduces a Reverse Training Scheme. We do not train on the target device's data (except for the classification head update). Instead, we use a few unlabeled target traces solely as "queries" to search the original profiling pool. This allows us to find profiling traces that mimic the target's signal characteristics (via KL-divergence and Feature distance). We train the model on profiling data, not target data. This enables attacks with minimal target interaction, which Dual-Leak cannot achieve.
>
> **W2. Theoretical Justification.**
> and **W3. Theorem 1 Justification.**
>
> We thank the reviewer for the rigorous examination of our theoretical justifications. We agree that the formal statement of Theorem 1 and its interpretation require greater precision to strictly align with the practical implementation of ActSCA. While the strategies involve hyperparameters, they are not ad-hoc; they correspond to minimizing specific components of error. For example, $D_1$ minimizes the covering radius ($R$) of the input space. By using K-medoids and BMMS, we ensure that every point in the full dataset $D_P$ is close to a selected point in $D_1$. Theorem 1 relies on this $R$ to bound the loss difference.
>
> We agree with reviewer is correct that Theorem 1 is stated for a fixed subset $S$ (centroids). However, BMMS  is designed to iteratively reduce the covering radius $R$ by selecting points furthest from the current set. Since the bound in Theorem 1 is proportional to $R$, BMMS explicitly tightens this bound. We will clarify that BMMS is the mechanism for minimizing $R$. Regarding the zero-loss assumption, we agree this is a strong assumption for parametric models. In the revision, we will relax this to an $\epsilon$-loss assumption ($l(x_j, y_j) \le \epsilon$).  This retains the main insight that the generalization gap is controlled by the covering radius $R$ and the training error $\epsilon$ on the subset.
>
> **W4. Results Presentation.**
>
> Thank you for your comments. When presenting the result, we want to show every aspect of the results and not just the usual end results to give the audience a wider view of the results. We will take this into consideration and readjust the figure.
>
> **W5. Label Distribution in BMMS.**
>
> We are sorry for not include an assumption about the label distribution in $S$ since the set $S$ has uniform distribution of label in our experiment. Therefore, $D1$ will have an uniform distribution of label as well. We will add a figure that showed the label distribution of the original set $S$ in the appendices.
>
> **W6.Evaluation**
> and **Q4.Evaluation ...**
>
> We include the single-trace attack result on Kyber in Table 1 and AES using ASCAD dataset in Table 2. ActSCA still outperform Baseline with a single trace evaluation on both dataset.
>
> *Table 1. Mean-rank of ActSCA with small number of traces on Kyber*
> | #Trace   | 1     | 10    |
> |----------|-------|-------|
> | Baseline | 56.84 | 29.03 |
> | ActSCA   | 21.73 | 6.01  |
>
> *Table 2. Mean-rank of ActSCA with small number of traces on ASCAD*
> | #Trace   | 1     | 10    |
> |----------|-------|-------|
> | Baseline | 59.54 | 8.74 |
> | ActSCA   | 45.87 | 7.57 |
>
> Thank you for your suggestion. We will change the baseline results to bold. With this change, Phase C of ActSCA still demonstrate its effectiveness by outperforming baseline in 5 out of 6 cases.
>
> **W7. Notation.**
>
>  We will update our paper accordingly
>
> **W8. Typos**
>
> . We will update our paper accordingly
>
> *(To be continued...)*

---

> ### Author Response · Authors · 2025-12-03
> **Response to Reviewer 1PyF (Part 2)**
>
> **Q1. The distance metric on L179 is simply the Euclidean distance, although we wish to measure similarity between time-series data. Have the authors tried distance-like measures that are more appropriate for time-series data (e.g., dynamic time warping distance)?**
>
> We have experimented with several distance metrics as presented in Table 13. In fact, when developing the algorithm, we already employed Dynamic Time Warping (DTW). It shows some potential results when testing with small toy datasets. However, since the computational cost of DTW is extremely high, we decided to go with a simpler but faster Euclidean distance (running DTW on 200K traces with 600 points each can take up to 523 years with 0.431s per calculation). Luckily, the results are still good. One reason is that: in each phase of our algorithm, the data is chosen from two different perspectives: one is data perspective using distance measures, and one is model perspective using model outcomes. Hence, they can complement each other to overcome their weaknesses.
>
> **Q2.It seems ... filter extreme outliers?**
>
> We appreciate this insightful observation. We agree that if BMMS relied solely on distance maximization (i.e., when $\gamma \to 1$), it would indeed prioritize "pathological" outliers, potentially degrading model performance. We empirically observed this exact phenomenon in our ablation studies. As shown in Figure 13(A), the performance worsens significantly as the regulation parameter $\gamma$ approaches 1.0, where the method selects points based purely on distance. Thus, while explicit outlier filtering is a valid potential extension, our results suggest that the Balanced Max-Min strategy with already effectively neutralizes the negative impact of outliers while retaining the benefits of data diversity.
>
> **Q3. How crucial ...**
>
> We investigate the effect of $\gamma$ in Figure 13A and we observed that the attack performance is worse without label balance. The combination of both label balance and BMMS with $\gamma = 0.5$ yield the best result. Similarly, we also investigate the effect of $\alpha$ in Phase B in Figure 11H. $\alpha$ in range [0.25,0.75] provide more stable results so we fixed $\alpha$ at 0.5. The label distribution of $D_P$ is uniform e.g each key has similar number of traces.
>
> **Q5.The authors compare their ...**
>
> Methods like ZMUV (Zero-Mean Unit-Variance), MMD (Maximum Mean Discrepancy), and ADA (Adversarial Domain Adaptation) rely on aligning the statistical distributions of the source and target domains. To accurately estimate these statistics (e.g., variance, distribution density, or domain-invariant features), these methods fundamentally require a large pool of target traces. When constrained to a very small set (e.g., 10 traces), the statistical estimates become highly noisy and unrepresentative of the true target distribution. Because the alignment mechanisms in MMD and ADA are driven by these noisy estimates, they force the model to adapt to a "false" target distribution. This often corrupts the well-learned features from the source domain, leading to negative transfer where the adapted model performs worse than the non-adapted Baseline (which at least preserves the robust features learned from the large source pool).
>
> In contrast, ActSCA does not attempt to estimate the global distribution of the target device. Instead, it employs a Reverse Training Scheme where the few target traces act as queries to retrieve similar instances from the rich profiling pool. This "instance-based" approach is robust to small sample sizes because it relies on the quality of individual matches rather than the accuracy of global statistical estimates,

---

### Official Review · Reviewer_tgu2 · 2025-10-31

**Soundness:** 1
**Presentation:** 2
**Contribution:** 1
**Rating:** 2
**Confidence:** 5

**Summary:**

The paper presents Active Side Channel Analysis (ActSCA), a framework designed to enhance Deep Learning-based Side Channel Attacks (DL-SCA), which aim to extract sensitive information by analyzing physical signals from cryptographic devices. Traditional DL-SCA methods typically require large datasets and lack adaptability across different devices. ActSCA seems to address these limitations through two innovations: Active Training and Adaptive Attacking. The authors validate the effectiveness of their approach through experimental evaluations conducted on both local and public datasets.

**Strengths:**

(1) This idea is straightforward and can be easily understood.

(2) The experiments consider multiple machine learning models, with seemingly more rigorous evaluation methodologies than many previous papers in this area.

**Weaknesses:**

(1) My major concern with this paper is its lack of novelty. Since deep learning (DL)-based side-channel analysis (SCA) is not new, the contribution of this work appears to be quite limited. The authors appear to simply extend one of the existing active learning methods—uncertainty sampling—and make a minor modification to it (referred to as balanced uncertainty sampling) to generate a sample set that is then used to update the DL model. The main focus of this paper still lies in applying some very  basic ideas from the DL domain to the SCA domain to reduce training costs and improve portability—issues that have already been addressed in prior research over the past few years. I believe the authors can easily find many similar papers published in well-known venues such as CHES, ICCAD, DAC, and others. Therefore, this paper does not seem to significantly advance our qualitative understanding of this area within the security domain.

(2) The adversary (or threat) model that the authors aim to address is also somewhat confusing. For example, in a profiled attack, we typically have two devices: the profiling device and the target (or attacking) device. Do the authors assume that these two devices share similar or different architectures, or that they use the same or different AES implementations? Moreover, while this paper focuses on profiled attacks, such attacks are difficult to deploy in real-world scenarios. In practice, the attacker often cannot obtain a copy of or a similar device for profiling the DL model before applying it to the target device for secret key recovery.

(3) The experimental setup is also unclear. Do the authors use fixed keys or varied keys for testing?

(4) Finally, the writing needs significant improvement. In critical parts of the paper, it is hard to tell what the authors did in terms of experimentation and analysis, or what motivated the choices they made.

**Questions:**

Please refer to my comments for more details.

---

> ### Author Response · Authors · 2025-12-03
> **Response to Reviewer tgu2**
>
> **W1.My major concern...**
>
> We respectfully disagree with the assessment that the contribution is limited to a minor modification of uncertainty sampling. While we acknowledge that Deep Learning (DL) in Side-Channel Analysis (SCA) is an established field, ActSCA introduces a fundamental paradigm shift in how portability and data efficiency are handled, distinct from existing works in CHES, DAC, or ICCAD.
>
> Our contributions go significantly beyond "Balanced Uncertainty Sampling" (which is just one component of Phase B). The core novelties lie in the Reverse Training Scheme (Phase C) and the Data-Centric Regularization (Phase A), backed by theoretical proofs.
>
> In phase C, instead of training on target data, we use the target traces (which can be as few as one unlabeled trace ) solely as a "query" to search the original profiling pool for similar signal characteristics, then fine-tune the model on these selected profiling traces. This allows for zero-shot or few-shot adaptation without requiring any training data generation on the victim device, a practical breakthrough that existing "standard" DL-SCA methods do not offer.
>
> The reviewer mentions we essentially use uncertainty sampling in Phase B. However, standard uncertainty sampling suffers from "catastrophic forgetting" in SCA contexts. To deal with this problem, we introduce a cluster-based memory replay mechanism. When selecting new uncertain samples, we augment them with representative samples from their clusters. This is not a "minor modification" but a critical architectural choice that stabilizes the decision boundary in the noisy SCA domain.
>
> **W2. The adversary...**
>
> We assume the profiling and target devices share the same architecture (e.g., both are STM32F3 Cortex-M4 microcontrollers for the Kyber experiments). The core challenge we address is not architectural difference, but cross-device portability: even identical chips from the same manufacturer exhibit significant signal variations due to manufacturing tolerances and electrical noise, which often causes standard models to fail.
>
> We respectfully argue that profiled attacks are highly relevant in real-world scenarios, particularly for IoT and embedded security. For example, in mass-market scenarios (e.g., smart cards, IoT nodes), an attacker can easily acquire a device identical to the victim's (open market purchase) to act as the profiling device. The "difficulty" lies not in obtaining the device, but in the fact that the model trained on the copy does not transfer to the victim due to physical signal mismatch.
>
> **W3.The experimental setup is also unclear. Do the authors use fixed keys or varied keys for testing?**
>
> We use varied keys for all the experiments as we emphasize that single key evaluation e.g ASCAD fixed key dataset is biased.
>
> **W4.Finally, the writing...**
>
> Thank you for your comment. We will improve our writing accordingly.

---

### Official Review · Reviewer_SRPA · 2025-11-01

**Soundness:** 3
**Presentation:** 2
**Contribution:** 2
**Rating:** 4
**Confidence:** 3

**Summary:**

This paper presents ActSCA, a framework to train deep-learning based profiling attack models more effectively. The proposed method is mainly made with three phases. In phase A, it trained the model on a selective subset of data, In phase B, it iteratively trains the model on another selective subset of data that the model predict poorly with high uncertainty. In phase C, which is during the attacking time, it retains the model with another selective subset of data that have the most similarity to the data samples from the target device. Empirical results show that it achieves a better attack performance in terms of mean rank and top-k accuracy compared to the baseline.

**Strengths:**

1. The empirical results shown in Fig. 3 demonstrated a large margin of improvement using the proposed method.
2. Over all, this paper is well structured and easy to follow.

**Weaknesses:**

1. I don't think the math in session 2.1 helps to prove any point. In the paper, a repetitive statement is that "using the full dataset for training suffers from overfitting". But it is lack of evidence to show it is indeed the reason of worse performance, or explanation or discussion around why training on a small subset helps to avoid overfitting. Intuitively, training on less data is more likely to overfit (to that smaller subset of data), and it is not clear why it avoid overfitting on the full data and why carefully choosing the subset of data is important.
2. There are criteria in phase A and phase B on how to choose the subset of data. Although intuitively they all makes sense, the paper is lack of empirical evidence that using those criteria leading to a better performance compared to the baseline. For instance, it would be interesting to see what if in phase A we randomly choose some samples. What is the performance difference between randomly sampled subset and the carefully chosen subset. This kind of ablation study is important to demonstrate the importance of the proposed tricks.
3. Overall the contribution is not that significant. The methods in the paper is more like tricks that principled approaches.
4. The graphs are blurred and the font size is too small to read.
5. I feel the experiment results can be better presented in tables than in figures.

**Questions:**

1. As already discussed in "Weakness", why training on a subset of data helps to avoid overfitting?
2. How much more computation is introduced with the proposed method (as it needs to sample the data) compared to the baseline when obtaining the experiment result?

---

> ### Author Response · Authors · 2025-12-03
> **Response to Reviewer SRPA**
>
> **Q1. As already discussed in "Weakness", why training on a subset of data helps to avoid overfitting?**
>
> **W1. I don't think the ... data is important.**
>
> While it is true that smaller datasets can be easier to overfit, the key idea behind using $D_1$ is that it is not a random subset but a carefully selected and diverse representation of the entire data pool $D_P$. Hence, $D_1$ captures the underlying structure of the data distribution while reducing redundancy. This encourages the model to learn more generalizable features, particularly in the early training phase.
>
> **W2. There are criteria in ... the proposed tricks.**
>
> We did include the ablation study for this choice of design. Please refer to Figure 7 for the performance differences between multiple subset selection strategies include Random, Minmax Balance, JTT, CORESET,... Similarly, we include ablation studies for the design choice of Phase B with different selection strategies in Figure 6.
>
> **W3. Overall the contribution is not that significant. The methods in the paper is more like tricks that principled approaches.**
>
> We believe the contribution is significant because ActSCA addresses the two big challenges in practical Side-Channel Analysis: portability and data efficiency, which standard principled approaches fail to solve effectively.
>
> A major contribution is our novel reverse training scheme. Unlike existing transfer learning methods (e.g., Dual-Leak) that require large, labeled datasets from the target device (which is often impractical), ActSCA exploits the existing profiling pool to adapt to the target. This paradigm shift allows for highly practical, label-free cross-device attacks, which addresses a critical gap in the literature.
>
> The method yields massive performance gains, outperforming current SOTA methods (such as MDMSD and ADA) by orders of magnitude (up to 300x improvement in Mean Rank).
>
> Furthermore, unlike "tricks" that are often specific to one dataset, ActSCA is a generic framework validated on both Post-Quantum (Kyber) and classical (AES) cryptography, including protected implementations.
>
> **W4. The graphs are blurred and the font size is too small to read.**
>
>  Thank you for your comment. We will improve the figure readability by resizing the graph and font size. The content in the paper is really dense and sometimes it is quite difficult to fit all content in 9 pages.
>
> **W5. I feel the experiment results can be better presented in tables than in figures.**
>
> Thank you for your comment. We utilized figures in the main text to illustrate the stability and convergence speed of the attacks, which are dynamic properties difficult to capture in a single table. However, to address your concern regarding precise comparison, we have ensured that detailed numerical breakdowns (e.g., Mean Ranks over 10 and 100 runs) are provided in Tables 7, 8, and 15.
>
> **Q2. How much more computation is introduced with the proposed method (as it needs to sample the data) compared to the baseline when obtaining the experiment result?**
>
> Our method introduces negligible computational overhead compared to the significant efficiency gains from data reduction, resulting in a net computational saving over the baseline. While ActSCA requires an initial, one-time computation for sampling the data (clustering and uncertainty calculation), the subsequent benefit of training on a smaller, highly informative subset dramatically outweighs this cost. Specifically, ActSCA ultimately uses only 66.1% of the full training pool, is much more efficient. The training runtime for ActSCA Phase B is 40s per epoch, which is nearly 50% faster than the 84s per epoch reported for the baseline CNNC model trained on the full dataset, leading to a substantial reduction in the overall time and energy required to obtain the final experiment result.

---

### Official Review · Reviewer_hXbE · 2025-11-01

**Soundness:** 2
**Presentation:** 3
**Contribution:** 2
**Rating:** 2
**Confidence:** 3

**Summary:**

Active SCA (ActSCA) is a framework for boosting performance of any base DLSCA model. Instead of large training data in the profiling stage, it actively selects subsets of training data and iteratively refines the model to avoid overfitting. In the attack phase, ActSCA uses the training data and a small amount of target device's traces to construct a separate attack model that works well for specific target devices.

**Strengths:**

- **Relevant topic of portability for SCA.** The paper considers portability issues in (DLSCA) where the trained models do not generalize enough to another device (clone or similar).
- **Extensive experiments performed and reported.** The authors consider different keys, devices, crypto systems and attack methods, while also performing multiple runs for statistical significance.
- **Better performance than compared methods.** The paper reports significant improvements over many methods for two crypto systems.

**Weaknesses:**

- **Design choices are not always clear.** For example, the Phase A selection method involves clustering, however it is not clear why this specific clustering is good specifically for SCA traces. Traces further from a centroid could be more noisy, leaking less information, allowing the method to still select "bad samples", as the authors also discuss in the paper. In Phase B, again, choosing samples based on uncertainty seems to also lead to selection of possibly some outlier/noisier traces.
- **Evaluation has some flaws.** Guessing entropy is more commonly used due to being more representative and stable than a key rank metric, while mostly key rank is reported in this work. SoTA on ASCAD dataset does not seem to be SoTA as there are works much successful than those reported here, where GE of 1 is achieved with a small amount of traces. Baseline is unclear as it is the proposed method using the complete training dataset while authors do not report how the phases are adapted to that change and, there should be some, since most of the method is about selecting samples from dataset.
- **The method might not be as generic as claimed.** The results on different model architectures vary significantly (based on results in Table 1. in Appendix.)
- **One of the issues authors are trying to solve in this work is not a relevant research question**, and it has different known ways to handle it. In particular, this is the question: "can we improve the model performance by carefully exploiting a subset of data rather than asking for more data?". There are more common methods, such as data preprocessing, removing outliers, regularization methods and many more that work on improving model's generalization, performance, etc. I'm not convinced this is an issue requiring such an elaborate method for solving it.

**Questions:**

- What makes clustering a good method for selecting representative SCA traces? How can you avoid selection of "bad samples"?
- In Phase B, why do you choose based on uncertainty?
- Why is key rank the main metric in this work?
- Table 12 reports results on ASCAD dataset, standard benchmark for DLSCA, and it is known that this dataset can be broken with very few traces, thus the results reported are not representative. How come SoTA for ASCAD does not have better performance?
- You should explain in more detail your Baseline method. Since it is using the complete dataset, it is unclear how the steps of your proposed methods are modified.
- The results on different model architectures vary significantly (based on results in Table 1. in Appendix.). Why is that and does that not show that the method is somewhat dependent on the architecture?
- Training data from the target/attack device is not a requirement in the standard DLSCA framework (as you also state in the abstract). Training data is assumed from a clone device, while from the target data, measurements (without labels) are collected. So, it is unclear why is this fact so emphasized in this work as it is not a novel thing or an improvement over other work that the training data is not used. It also confuses reader to think that training data is normally used in DLSCA which is not true. Thus, perhaps it would be better to focus on what is it that you do in the attack phase to allow portability to different devices, while removing this repeated emphasis on attack traces not being part of training.
- One of the motivations for Phase A (using subset of training data) is overfitting. Overfitting can be reduced/mitigated with regularization, hyperparameter tuning, and other already well-established methods in ML domain. So, why is this method better than these other techniques?
- Large datasets require increased computation costs, but this method also seems to require a lot of computational power for all the phases and the procedures in each of them. In appendix you provide some information on runtime, but Phase A is not shown but is a part of your proposed method, making this unfair comparison. Can you provide more runtime analysis?
- It is not clear which level of portability are you considering until the experiments section. With the level of portability, I mean that you can look at the portability between identical devices where the difference comes from manufacturing process, or you can be looking at actual different hardware or algorithm running. So, can you give some indication on which portability level are you considering earlier in the text, as it will also help understand the proposed method better.


Additional comments/questions (less relevant)
- In appendix there is mention of 50K validation set. How is that one used in your proposed attack method?
- The profiling dataset of 200K samples, is it collected with the 300 different keys as well, or a single one?
- Not clear why the name reverse training scheme.
- KNN not defined.
- Figure 4 - What are the axis?
- "Unused training data will be retained in the training pool and used as or unseen data..." - used as what?
- What are CNNC, MDMSD, ZMUV abbreviations of?
- Figures should be closer to the text referencing them.
- hyper-parameter turnings?
- Table 11 and text about that table report slightly different numbers (118-119, 86.2-87.2)

---

> ### Author Response · Authors · 2025-12-03
> **Response to Reviewer hXbE (Part 1)**
>
> **W1. For example, the Phase A selection method involves clustering, however ... possibly some outlier/noisier traces.**
>
> **Q1. What makes clustering a good method for selecting representative SCA traces? How can you avoid selection of "bad samples"?**
>
> It is a great question. The key idea of using clustering is that a sample lying very deep inside the cluster will be the most representative for the whole cluster. Hence, choosing it would be a good choice (possibly more similar to the rest, less noisy, etc..). By this way, theoretically any clustering algorithm can be used. However, some like k-Means or k-Medoids are particularly suitable due to their center of mass concepts around the centers and centroids. Other methods like DBSCAN can create arbitrary shape clusters, hence causing difficulties when choosing their center samples. In our paper, we chose k-medoids as the main method. We also demonstrated performances using k-means in Table 5 of the appendix. There are no clear differences between k-Means and k-Medoids which is not a surprise since centroid or center samples are quite close in concept and they are only just a starting point of our framework. There are many proposed key technical contributions in 3 Phases A, B and C to further boost the performance.
>
> We agree with the reviewer that samples far away from centroids *may be a less important* one. However, it is also not entirely true. They also *may be a very important sample* (for example, belong to a minority class or contain information in some very specific cases). Hence, we proposed to enhance the diversity by choosing samples that are scattered around the whole data space to supplement the centroid ones, keeping an eye on label imbalance, a critical problem which can happen during data selection as stated via Eq. 2. In short, distances to centroids are not the sole condition but label distributions in the current selected set S. We experimentally proved in Figure 7A,B  that our centroid + diversity extension scheme leads to significant performance improvements compared to various schemes like random sampling or distance only. This showed that though we cannot avoid bad samples being involved (and we believe there is hardly a perfect way, even the term bad samples here does not mean the sample is noisy but they contribute less to the over performance of models), most of the samples we selected actually help to improve the performance.
>
> We also need to note that all of the data has been cleaned, noisy data has been removed during the data collection process. Though there is no way to fully ensure perfect data, very bad samples (for example those with extreme magnitude) are more likely to be excluded.
>
> **Q2. In Phase B, why do you choose based on uncertainty?**
>
> We have tried many different strategies (some of them are inspired by Active Learning such as Uncertainty or Margin, though we need to highlight that our Active selection scheme is entirely different to traditional active learning, where a pool of unlabeled data is used but we have a pool of labeled data, a key difference in the concept, that lead to our special selection scheme in Eq. 5 with label distributions). Among them, we see that Uncertainty scheme, where the labels of samples are less certain and they need to be focused to clarify their label, lead to better performance. So, we chose them. But there may be a better way, which can be discovered in the future. We also note that our selection process is not purely based on uncertainty but on the label distribution and clustering structure as we stated in Eq. 5. We studied performances of different strategies in Figure 6.
>
> Similar to Phase A, there is no way we can be 100% sure that bad samples still slipped through. However, the experiments showed that the selection scheme is effective via significant performance enhancement as we can see in Figure 6 too.
>
> **Q3.Why is key rank the main metric in this work?**
>
> **Q4.Table 12 reports results on ASCAD dataset..better performance?**
>
> **W2. Guessing entropy is more commonly used due to ... selecting samples from dataset.**
>
> Both Mean-rank (or can be called key rank) and GE are two main metrics used in SCA researches. In our work, we use key rank as the main metric due to the limited number of traces on the attacking device (<100 per key). A small number of samples provides an unstable, potentially misleading approximation of the true expected rank over the complete key space when using GE as a metric. Upon Key rank, we also included GE and Median Mean Ranks when working with ASCAD dataset since the number of attack traces per key is more abundant. The results of GE can be found in Table 4 in the Appendix, demonstrating the same conclusions with Mean-ranks.
>
> *(To be continued…)*

---

> ### Author Response · Authors · 2025-12-03
> **Response to Reviewer hXbE (Part 2)**
>
> In ASCAD dataset, we have two variants ASCAD_R (Random key) and ASCAD_F (Fixed key). ASCAD_R is much harder compared to ASCAD_F and is our target in the paper. Most of these baselines are built to adapt to cross-device scenario and ASCAD is not a cross-device dataset so the results might not be inline with work that built a single device model such as Estranet [F]. Furthermore, such baselines usually requires a huge amount of traces on the target device, e.g 40k traces needed for ADA as they reported in the paper. Also, most papers reported the results of ASCAD_F and do not include results on ASCAD_R dataset. For example, MDMSD does perform experiments on ASCAD but they only include GE differences for each layer and does not report the GE result on the full model. Other methods such as ZMUV, ADA and MMD do not report the result on ASCAD_R.  In DL-SCA, people tend to heavily tailored common models like MLPs or CNNs to achieve better performance via different learning hyper-parameters, etc. So, some models with the same MLP or CNN architectures like us but with different performance may exist (*we appreciate very much if the reviewer would help us by pointing out those that we missed*). However, regardless of the results they reported. Assume that there is a specifically tailored model $M$ with SOTA performance on an arbitrary dataset $D$, we can employ $M$ as a base model in ActSCA. And as a generic framework, ActSCA is expected to boost performance of that model $M$ on $D$ significantly.
>
> **Q5.You should explain in more detail your Baseline method. Since it is using the complete dataset, it is unclear how the steps of your proposed methods are modified.**
>
> Baseline is the base model we used inside ActSCA, which can be CNN, MLP, etc. (any existing DL models for SCA can be used here. In the paper, we chose some most common architectures). **Our target is to prove that ActSCA can boost performance of any employed models**. Our baseline method follows the traditional approach of training ML models in the literature. We use CrossEntropy Loss with RMSpropOptimizer and train the model until convergence.  Please refer to Section C in the appendices for more details.
>
>
> **Q6. The results on different model architectures vary significantly (based on results in Table 1. in Appendix.). Why is that and does that not show that the method is somewhat dependent on the architecture?**
>
> The model does not perform well on Transformer compared to CNN and MLP.This degradation in performance may stem from the mismatch between the Transformer’s design and the nature of side-channel traces. Another reason might be the Transformer architecture of Estranet might not be suitable for all dataset since they are fine-tuned on longer trace length (10K and 40K compared to usual 1400 on ASCAD and 600 on Kyber), which is beneficial for the self-attention mechanism of Estranet. However, the performance we reported there is the raw performance of Transformer, not performance of ActSCA. When using ActSCA, we expect that our framework will significantly improve the results of Transformer as we demonstrated with many other architectures such as MLP, CNN and Estranet. We skipped Transformer in ActSCA study since its performance is far worse than others.
>
> **Q7. Training data from the target/attack device is not a requirement in the standard DLSCA framework (as you also state in the abstract). Training data is assumed from a clone device, while from the target data, measurements (without labels) are collected. So, it is unclear why is this fact so emphasized in this work as it is not a novel thing or an improvement over other work that the training data is not used. It also confuses reader to think that training data is normally used in DLSCA which is not true. Thus, perhaps it would be better to focus on what is it that you do in the attack phase to allow portability to different devices, while removing this repeated emphasis on attack traces not being part of training.**
>
> Training data from the target/attack device is not a requirement in the standard DLSCA but works focused on portability such as [D, E] need a huge amount of training traces from the target device to fine-tune their model. To that end, our methods do not need to collect any traces from the target devices to train attack models. Instead, it only needs to use a small number of unlabeled trace collected during the attacking stage. Hence, it is a significant advantage of the attack model for portability and makes ActSCA a very practical attacking method.
>
> *(To be continued…)*

---

> ### Author Response · Authors · 2025-12-03
> **Response to Reviewer hXbE (Part 3)**
>
> **Q8. One of the motivations for .. other techniques?**
>
> We appreciate the reviewer's suggestion regarding standard regularization techniques [e.g L2, Dropout]. While these methods reduce model complexity "blindly," our Phase A strategy offers a **data-centric regularization** that is specifically tailored to the Side-Channel domain.
>
> The model may "overfit" to the subset of data, but because this subset represents the most informative part of the signal, it prevents the model from overfitting to the noise of the entire known data space. In Phase A, ActSCA explicitly selects a "compact yet informative subset" ($D_1$) using k-medoids clustering and Balanced Max-Min Sampling (BMMS). This subset consists of centroids and diverse samples that capture the deterministic leakage signal rather than the random noise. By restricting the model to this subset, we force it to lock onto the representative structure of the leakage. Even if the model achieves near-zero loss on this subset (technically "overfitting" the subset), it is beneficial because it has learned the invariant features of the key leakage, avoiding the trap of fitting the stochastic noise present in the full dataset.
>
> **Q9. Large datasets require increased .. runtime analysis?**
>
> We thank the reviewer for highlighting the runtime comparison in the Appendix. We agree that a complete computational picture is important. However, the exclusion of Phase A from the runtime comparison table (Table 2) was intentional, driven by the operational distinction between one-time initialization (Phase A) and the iterative training/attacking phases (Phase B and C).
>
> Phase A  is a pre-processing step performed **once** on the profiling device. It uses clustering (K-medoids) to select an initial representative subset $D_1$2 Unlike model training, which requires hundreds of epochs, Phase A is executed a single time to initialize the data pool. Once $D_1$ is selected, it serves as the foundation for all subsequent model updates. Our primary research goal is to optimize the Attack Stage (Phase C) and the iterative refinement (Phase B), as these are the bottlenecks in realistic Cross-Device scenarios. As shown in Table 2, ActSCA reduces the runtime per epoch significantly (40s and 29s vs. 84s for CNNC) because it trains on smaller, selected subsets ($D_2$ and $D_3$) rather than the full pool. The computational cost of Phase A is effectively amortized over the subsequent phases. By spending time upfront to select high-quality representative data in Phase A, we significantly reduce the computational load in Phase B and C, allowing the attack model to converge faster and with less data.
>
> To address your request for specific analysis, we emphasise that Phase A relies on K-medoids clustering. While clustering 200,000 traces does incur a computational cost, we utilized optimized implementations (Fast K-medoids/PAM). In our experiments, this clustering process took approximately 30-60 minutes.
>
> In the final version, we will add a row or footnote to the runtime table explicitly stating the one-time execution time of Phase A to ensure transparency, while highlighting that this cost is offset by the significant speedups in the active training and attacking phases.
>
> **Q10. It is not clear which level of portability are you considering until the experiments section. With the level of portability, I mean that you can look at the portability between identical devices where the difference comes from manufacturing process, or you can be looking at actual different hardware or algorithm running. So, can you give some indication on which portability level are you considering earlier in the text, as it will also help understand the proposed method better**
>
> We thank the reviewer for pointing out this ambiguity. We agree that "portability" can refer to various scenarios (Cross-Device, Cross-Architecture, Cross-Implementation).
>
> In this work, we specifically consider Cross-Device Portability between identical hardware models. Our focus is on the scenario where the Profiling Device (PD) and the Target Device (TD) share the same microcontroller architecture (e.g., both are STM32F3), but are different physical instances. In this context, the domain shift arises from manufacturing variations (silicon differences), measurement setup discrepancies (probe placement, cabling), and environmental noise, rather than fundamental algorithmic or architectural differences.
>
> References:
>
> [A] Active Learning for Convolutional Neural Networks: A Core-Set Approach, ICLR, 2018.
>
> [B]SoK: Deep Learning-based Physical Side-channel Analysis
>
> [C]Side-channel analysis attacks based on deep learning network
>
> [D]Cross-device profiled side-channel attacks using meta-transfer learning. In DAC, 2021.
>
> [E]Dual-leak: Deep unsupervised active learning for cross-device profiled side-channel leakage analysis. In HOST, 2023.
>
> [F]Estranet: An efficient shift-invariant transformer network for side-channel analysis." CHES, 2024

---

### Meta-Review · Area_Chair_Xhiz · 2026-01-06

**Summary:**

While the problem studied is interesting, the reviewers raised a large and broad set of concerns, including readability, motivation, soundness, correctness of the stated (theoretical) foundations, experimental validation, and significance of the methodological contributions. Overall, this work and manuscript does not seem ready for publication.

**Reviewer Concerns:**

The authors provided detailed responses to the reviewers, elaborating on the choices and motivation, and acknowledging some of the weaknesses. However, the concerns regarding significance of the methodological contributions remain. Similarly, supporting evidence for the claims provided by the experimental validation were unconvincing. The reviewers provided an extensive set of comments that the authors could use to revise the paper and improve the work presented.

**Reviewer Scores:**

All reviewers voted for rejection with a relatively high confidence. They provided constructive comments to the authors, who provided detailed responses. Due to the many weaknesses of this paper, including presentation and soundness, and the general consensus among the reviewers that the work is not meeting the acceptance bar, I do not expect that the scores would have been raised post rebuttal to warrant acceptance.

---

### Decision · Program_Chairs · 2026-01-26

Reject